# ILA-DA: Improving Transferability of Intermediate Level Attack with Data Augmentation

**Chiu Wai YAN, Tsz-Him CHEUNG & Dit-Yan YEUNG**
Department of Computer Science and Engineering
The Hong Kong University of Science and Technology
Clear Water Bay, Kowloon, Hong Kong
{cwyan, thcheungae}@connect.ust.hk, dyyeung@cse.ust.hk

## Abstract

Adversarial attack aims to generate deceptive inputs to fool a machine learning model. In deep learning, an adversarial input created for a specific neural network can also trick other neural networks. This intriguing property is known as black-box transferability of adversarial examples. To improve black-box transferability, a previously proposed method called Intermediate Level Attack (ILA) fine-tunes an adversarial example by maximizing its perturbation on an intermediate layer of the source model. Meanwhile, it has been shown that simple image transformations can also enhance attack transferability. Based on these two observations, we propose ILA-DA, which employs three novel augmentation techniques to enhance ILA. Specifically, we propose (1) an automated way to apply effective image transformations, (2) an efficient reverse adversarial update technique, and (3) an attack interpolation method to create more transferable adversarial examples. Shown by extensive experiments, ILA-DA greatly outperforms ILA and other state-of-the-art attacks by a large margin. On ImageNet, we attain an average attack success rate of 84.5%, which is 19.5% better than ILA and 4.7% better than the previous state-of-the-art across nine undefended models. For defended models, ILA-DA also leads existing attacks and provides further gains when incorporated into more advanced attack methods. The code is available at https://github.com/argenycw/ILA-DA.

## 1 Introduction

Recent studies (Szegedy et al., 2013; Goodfellow et al., 2015) showed that deep neural network (DNN) models are vulnerable to adversarial attacks, where perturbations are added to the clean data to fool the models in making erroneous classification. Such adversarial perturbations are usually crafted to be almost imperceptible by humans, yet causing apparent fluctuations in the model output. The effectiveness of adversarial attacks on deep learning models raises concerns in multiple fields, especially for security-sensitive applications.

Besides being effective to the victim model, adversarial attacks are found to be capable of transferring across models (Papernot et al., 2016). One explanation for this phenomenon is the overlapping decision boundaries shared by different models (Liu et al., 2017; Dong et al., 2018). Such behavior not only aggravates concerns on the reliability and robustness of deep learning models, but also enables various black-box attacks which leverage the transferring behavior, such as directly generating attacks from a source (or surrogate) model (Zhou et al., 2018) or acting as a gradient prior to reduce the number of model queries (Guo et al., 2019).

Intermediate Level Attack (ILA) is a method proposed by Huang et al. (2019) to fine-tune an existing adversarial attack as a reference, thereby raising its attack transferability across different models. Formulated to maximize the intermediate feature map discrepancy represented in the models, ILA achieves remarkable black-box transferability, outperforming various attacks that are directly generated (Zhou et al., 2018; Xie et al., 2019a). On the other hand, many of the transfer-based attacks empirically show that simple image transformations, including padding (Xie et al., 2019a), image translation (Dong et al., 2019), and scaling (Lin et al., 2020), are effective in strengthening the trans-

Clean Image        I-FGSM        I-FGSM + ILA-DA

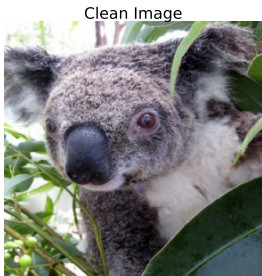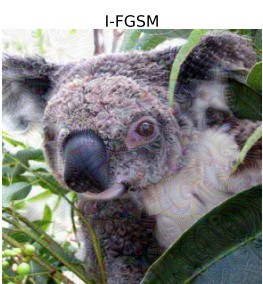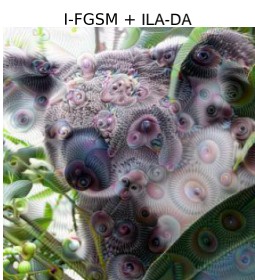

Figure 1: Visualization of the generated images among: clean image, I-FGSM, and I-FGSM + ILA-DA (Ours), with the perturbation budget $\epsilon = 16/255$ (0.063).

ferability of an adversarial attack. Similar ideas can be applied to the ILA framework, which takes three input images as references, including a clean image, a reference adversarial image, and the updated image from the previous iteration. This motivates us to advance the current transfer-based ILA attack by promoting reference diversity using three carefully-designed augmentation techniques.

In this paper, we revisit ILA and propose ILA-DA, which employs three novel augmentation techniques to advance ILA attack transferability across different model architectures. First, we develop an automated way to apply effective image transformations for more diverse augmented references. Unlike recent research on Automated Data Augmentation (AutoDA) (Cubuk et al., 2019; Li et al., 2020c) which constructs useful augmented data for model training, our approach applies learnable augmentations to create more powerful adversarial examples. Second, we propose an efficient augmentation method, called the reverse adversarial update, to create new augmented references by utilizing adversarial noises. Nevertheless, ILA-DA forms new augmented references by interpolating images in the direction that leads to larger intermediate feature discrepancies. Through fine-tuning simple baseline attacks, such as the Iterative Fast Gradient Sign Method (I-FGSM), we show that ILA-DA outperforms state-of-the-art transfer-based attacks on both defended and undefended models by large margins. Visualizations of some adversarial examples created by ILA-DA are illustrated in Figure 1 and Appendix O.

Our main contributions can be summarized as follows:

- We propose ILA-DA, which applies three novel augmentation techniques, including automated data augmentation, reverse adversarial update and attack interpolation, to considerably strengthen the current ILA attack.

- We demonstrate that ILA-DA outperforms various ILA attacks and other state-of-the-art approaches. On ImageNet, we attain an average attack success rate of 84.5%, which is 19.5% better than ILA and 4.7% better than the previous state-of-the-art across nine undefended models.

- We show that ILA-DA with simple I-FGSM attack references can exceed state-of-the-art attacks on six defended models. We also find that incorporating ILA-DA into existing attacks can further increase their attack transferability.

## 2   RELATED WORKS

White-box and black-box attacks are two common threat models used in adversarial attack research. The white-box setting assumes that the attacker has access to the victim model's internal state, including its gradient, parameters, training dataset, etc. The black-box setting, on the other hand, only allows model queries without accessing any subsequent model information. While more variations such as the grey-box and no-box settings exist, they are generally not considered in this work.

Typical white-box attacks exploit the gradient of the model to generate adversarial examples. The Fast Gradient Sign Method (FGSM) (Goodfellow et al., 2015) generates attacks by adding the signed gradient of the loss with respect to the image back to the image, obtaining a higher loss and possibly incorrect prediction. The Iterative Fast Gradient Sign Method (I-FGSM, also known as BIM) (Ku-

rakin et al., 2017a) performs FGSM iteratively while clipping the difference between the adversarial example and the original image within a perturbation budget. Projected Gradient Descent (PGD) (Madry et al., 2018) first initiates a random start within the perturbation budget and runs I-FGSM to update the attack.

Under the black-box setting, the model information can be estimated by repeated model queries with varying inputs (Chen et al., 2017; Brendel et al., 2018). Recently, there are multiple attempts to reduce the number of model queries (Bhagoji et al., 2018; Ilyas et al., 2018; Li et al., 2020a). However, in order to provide an accurate estimation to the model information, the required number of model queries is still abnormally large, causing the attacks to be detected by real-world systems easily. This turns the spotlight back to transfer-based attacks, which create adversarial examples through black-box transferability instead of repeated attempts.

## 2.1 TRANSFER-BASED ATTACKS

Since strong gradient-based white-box attacks usually do not exhibit high transferability (Kurakin et al., 2017b), notable efforts have been devoted to improve transferability apart from increasing the attack strength. An observable direction of the previous works is to incorporate carefully-designed data augmentation techniques into existing attacks.

**MI-FGSM.** Dong et al. (2018) first introduced the Momentum Iterative Fast Gradient Sign Method (MI-FGSM) to incorporate momentum to stabilize the update direction in iterative attacks, resulting in more transferable attacks.

**DIM.** The Diverse Input Method ($DI^2$-FGSM, or DIM) (Xie et al., 2019a) applies random resizing and zero-padding with a certain probability to the image in each iteration of I-FGSM.

**TIM.** Dong et al. (2019) pointed out the discretion of attention map between the defended and undefended models and proposed the Translation-Invariant Method (TIM) by attacking an ensemble of translated images. To reduce the computational overhead, it was shown that kernel convolution can be applied to the gradient to achieve similar effects.

**SIM and NI-FGSM.** Lin et al. (2020) introduced the scale-invariant property of DNN and proposed the Scale-Invariant Method (SIM) to attack an ensemble of images with a scaling factor of $0.5$ on the pixel values. In the same paper, the Nesterov Iterative Fast Gradient Sign Method (NI-FGSM) was introduced, aiming to escape from poor local maxima with the Nesterov accelerated gradient.

**Variance Tuning.** Wang & He (2021) proposed variance tuning. For each iteration, the gradient is adjusted with an expectation of gradient sampled from the image's neighborhood. Combining variance tuning with the composition of previous methods, denoted as the Composite Transformation Method (CTM), they formed the attack VMI-CT-FGSM and VNI-CT-FGSM, resulting in one of the strongest transfer-based black-box attacks on defended models. Following the convention, we will name this series of work the CTM family. A table matching the names and the alias of the CTM family is shown in Appendix I.

**Admix.** Wang et al. (2021) recognized the effectiveness of input transformation in attack transferability and proposed Admix. Inspired by Mixup (Zhang et al., 2018), Admix computes the gradient on a set of mixed images with different categories, which can be viewed as a special case of SIM, but with higher empirical contribution to transferability.

**SGM and LinBP.** Unlike the CTM family, another type of works successfully raise the attack transferability by exploiting CNN model architectures. Wu et al. (2020) found that attack transferability can be enhanced by scaling up the gradient in the skip connection and referred to the technique as Skip Gradient Method (SGM). Guo et al. (2020) revised the linear property of DNN models and proposed LinBP to propagate backward without considering the non-linear layers.

## 2.2 INTERMEDIATE LEVEL ATTACK FOR FINE-TUNING ADVERSARIAL EXAMPLES

The aforementioned works generate adversarial examples by maximizing the loss at the logits output. Nevertheless, some works leverage the intermediate feature of the CNN models. Activation Attack (Inkawhich et al., 2019) proposes to perturb the image such that its activation is closer to that of the target class. Feature Distribution Attack (FDA) (Inkawhich et al., 2020) perturbs the layer-

wise and class-wise feature distribution, which is determined by a set of auxiliary binary classifiers. Different from the others in which the attack is directly generated, ILA (Huang et al., 2019) uses a projection loss to fine-tune an existing reference attack.

Consider an input image $\mathbf{x}$, an existing adversarial attack $\mathbf{x}'$, a model $F$, and a function $F_l$ that outputs the feature maps at layer $l$ of the model. The original ILA projection loss proposed by Huang et al. (2019) minimizes the dot product:

$$L(\mathbf{x}, \mathbf{x}', \mathbf{x}'') = -\Delta \mathbf{y}_l'' \cdot \Delta \mathbf{y}_l' \tag{1}$$

where $\Delta \mathbf{y}_l'$ and $\Delta \mathbf{y}_l''$ are two vectors defined as follows:

$$\Delta \mathbf{y}_l' = F_l(\mathbf{x}') - F_l(\mathbf{x}) \tag{2}$$

$$\Delta \mathbf{y}_l'' = F_l(\mathbf{x}'') - F_l(\mathbf{x}) \tag{3}$$

This is equivalent to maximizing $\Delta \mathbf{y}_l'' \cdot \Delta \mathbf{y}_l'$. The idea behind the formulation is to increase the norm without sacrificing the perturbation direction that causes misclassification. Due to the constraint of the perturbation budget, there is hardly room for fine-tuning in the image space. Instead, projection maximization is carried out in the feature space, specifically at layer $l$ of the model.

Based on the same framework, Li et al. (2020b) proposed another formulation of ILA: $\max_{\mathbf{x}''} \left( F_l(\mathbf{x}'') - F_l(\mathbf{x}) \right)^\mathsf{T} \mathbf{w}^*$, where $\mathbf{w}^*$ is a pre-computed parameter vector that directly maps the intermediate-level discrepancies to predict the adversarial loss, skipping the remaining model layers. One notable trick of the design is that the computation of $\mathbf{w}^*$ involves every feature map discrepancy $F_l(\mathbf{x}_t') - F_l(\mathbf{x})$ in each iteration $t$ during the generation of $\mathbf{x}'$. We refer to this attack as ILA++ in the rest of this paper.

## 3 OUR METHOD

Based on what we have observed from the previous work, adversarial attacks tend to exhibit higher transferability when more references are involved. The increase of references can be achieved by data augmentation and reusing the temporal information over multiple iterations. From Equation (1), all three ILA input references $\mathbf{x}$, $\mathbf{x}'$ and $\mathbf{x}''$ are images, suggesting the possibility of applying image augmentation like that for representation learning to improve model generalization.

One natural way to augment the images is to apply image transformation to the input before retrieving the intermediate feature maps. Consequently, the function $F_l$ can be substituted by

$$F_l'(\mathbf{x}) = F_l(T(\mathbf{x})) \tag{4}$$

where $T$ can be any image transformation function.

Different from conventional data augmentation that generalizes model representation, our task is to improve attack transferability, which can be interpreted as generalizing an adversarial example to succeed in attacking more models. To achieve the goal, we propose three augmentation techniques: automated data augmentation, reverse adversarial update, and attack interpolation. The algorithmic details for applying these augmentation techniques in ILA-DA are depicted in Algorithm 1. The clip function is to ensure that the first argument is within the range defined by the second and third arguments.

### 3.1 AUGMENTATION LEARNING

Data augmentation is an effective technique to improve model generalization. However, it is non-trivial to select the most effective augmentation method for a specific task and dataset. As an Automated Data Augmentation approach, AutoAugment (Cubuk et al., 2019) uses reinforcement learning to search for the optimal augmentation policy for a dataset. With similar motivation, we propose an automated way to learn effective image transformations that lead to a more vigorous ILA attack.

We model the augmentation selection process $T_\mathbf{z}$ as sampling from a distribution $p_{\boldsymbol{\alpha}}(\mathbf{z})$ with the learnable parameter vector $\boldsymbol{\alpha}$ in Equation (5):

$$T_\mathbf{z}(\mathbf{x}) = \sum_{i=1}^{k} \mathbf{z}_i \varphi_i(\mathbf{x}), \ \ \mathbf{z} \sim p_{\boldsymbol{\alpha}}(\mathbf{z}) \tag{5}$$

---

**Algorithm 1** ILA-DA

---

**Require:** $\mathbf{x}, \mathbf{x}', F_l$, perturbation budget $\epsilon$, step size $a, \eta$, number of iterations $n$, a set of transformations $\{\varphi_i\}_{i=1}^k$, augmentation parameter $\boldsymbol{\alpha}$

Initialize $\boldsymbol{\alpha}$

$\mathbf{x}'' \leftarrow \mathbf{x}'$

**for** $i \leftarrow 1$ to $n$ **do**

    $\mathbf{x}_{\text{rev}} \leftarrow 2\mathbf{x} - \mathbf{x}'$                                     ▷ reverse adversarial update

    $\mathbf{z} \sim p_{\boldsymbol{\alpha}}(\mathbf{z})$

    $\Delta\mathbf{y}'_l \leftarrow F_l(T_{\mathbf{z}}(\mathbf{x}')) - F_l(T_{\mathbf{z}}(\mathbf{x}_{\text{rev}}))$

    $\Delta\mathbf{y}''_l \leftarrow F_l(T_{\mathbf{z}}(\mathbf{x}'')) - F_l(T_{\mathbf{z}}(\mathbf{x}_{\text{rev}}))$

    $\mathbf{x}'' \leftarrow \mathbf{x}'' - a \, \text{sign}(\nabla_{\mathbf{x}''}(-\Delta\mathbf{y}''_l \cdot \Delta\mathbf{y}'_l))$

    $\mathbf{x}'' \leftarrow \text{clip}(\mathbf{x}'', \mathbf{x} - \epsilon, \mathbf{x} + \epsilon)$

    $\mathbf{x}'' \leftarrow \text{clip}(\mathbf{x}'', 0, 1)$

    $\lambda \leftarrow \dfrac{\|\Delta\mathbf{y}''_l\|_2}{\|\Delta\mathbf{y}''_l\|_2 + \|\Delta\mathbf{y}'_l\|_2}$

    $\mathbf{x}' \leftarrow \lambda\mathbf{x}'' + (1-\lambda)\mathbf{x}'$                             ▷ attack interpolation

    $\boldsymbol{\alpha} \leftarrow \boldsymbol{\alpha} - \eta\nabla_{\mathbf{x}''}(-\Delta\mathbf{y}'_l \cdot \Delta\mathbf{y}'_l)$               ▷ augmentation parameter update

**end for**

**return** $\mathbf{x}''$

---

Sampled from $p_{\boldsymbol{\alpha}}(\mathbf{z})$, $\mathbf{z}$ is a one-hot vector that indicates the selection of the augmentation from a set of $k$ transformations $\{\varphi_i\}_{i=1}^k$. The overall goal is to maximize the ILA objective in Equation (1) using the augmentation sampled from $p_{\boldsymbol{\alpha}}(\mathbf{z})$. We use the Gumbel-Softmax reparameterization trick to relax the discrete augmentation distribution to be continuous (Jang et al., 2017). In so doing, we can utilize stochastic gradient descent, with a learning rate parameter $\eta$, to optimize the objective. The technique is also adopted in neural architecture search (Xie et al., 2019b) and AutoDA (Li et al., 2020c). With the reparameterization trick, we can soften $\mathbf{z}_i$ as:

$$\mathbf{z}_i = f_{\boldsymbol{\alpha}_i}(\mathbf{g}_i) = \frac{\exp\left((\log\boldsymbol{\alpha}_i + \mathbf{g}_i)/\tau\right)}{\sum_{j=1}^k \exp\left((\log\boldsymbol{\alpha}_j + \mathbf{g}_j)/\tau\right)}, \tag{6}$$

where $\mathbf{g}_i = -\log(-\log(\mathbf{u}_i))$ is the $i$-th Gumbel random variable, $\mathbf{u}_i$ is a uniform random variable and $\tau$ is the temperature parameter of the softmax function. In our implementation, we set $\tau = 1$ and sample from three common image transformation operations: translation, cropping, and rotation. It is worth noting that a perfect pixel alignment among $\mathbf{x}$, $\mathbf{x}'$ and $\mathbf{x}''$ is required in ILA-DA, as a tiny shift in the image pixel induces misalignment in the feature maps. In other words, translation, cropping and rotation are required to be applied identically to all three ILA references. Otherwise, the optimization of ILA ends up being trivial due to the distorted feature discrepancies.

## 3.2 REVERSE ADVERSARIAL UPDATE ON THE CLEAN EXAMPLE

Let us reconsider Equation (2). To perform ILA, we need to obtain the feature map discrepancies between the reference attack $\mathbf{x}'$ and the unperturbed image $\mathbf{x}$. Although $\mathbf{x}$ is correctly classified by the model, there is no information on how confident the model is and how well the input can represent the ground-truth class. In contrast, if $\mathbf{x}$ is not only correctly classified, but it is also done with a high confidence, the discrepancy of the feature maps is expected to intrinsically reflect more useful information about the attack.

In order to boost the confidence of $\mathbf{x}$, we propose to add a negative perturbation to it, which is exactly opposite to applying an adversarial attack. This operation is expected to decrease the loss of the classifier, making the classification task easier by emphasizing features that are more significant for the task. Thus, we also expect the input to highly activate the intermediate layers, providing a better guidance for image update in ILA. Our preliminary study on the effect of reverse adversarial update can be found in Appendix E, which verifies the benign behavior brought by the reverse adversarial update. With the reverse adversarial perturbation, as expected, the model gives a lower loss and higher confidence on the correct class.

Instead of crafting a new adversarial attack which would incur extra computational demand, we can extract the perturbation from the existing reference attack $\mathbf{x}'$, turning the transformation into

$$T_{\text{adv}}(\mathbf{x}) = \mathbf{x} - (\mathbf{x}' - \mathbf{x}) = 2\mathbf{x} - \mathbf{x}' \tag{7}$$

The notion of reverse adversarial update shares some similarities with DeepDream (Mordvintsev et al., 2015), in which an image is updated towards the models' interpretation of the target class. After the transformation, we look for the feature discrepancies caused by a bi-directional perturbation $\pm\epsilon \, \text{sign}(\nabla_{\mathbf{x}} J(\theta, \mathbf{x}, y))$, with the positive side being the adversarial attack and the negative side being the augmentation.

### 3.3 ATTACK INTERPOLATION ON THE REFERENCE ATTACK

Besides the clean image $\mathbf{x}$, the reference attack $\mathbf{x}'$ is another factor to remain unchanged throughout all the ILA iterations. Given the image transformation in Equation (4), it is unlikely to work with another image transformation unaligned with the other images. Similar to Section 3.2, we instead exploit the adversarial noise to be the augmentation.

According to the intuition of Best Transfer Direction (BTD) in Huang et al. (2019), the stronger the reference attack is, the better ILA performs. This inspires us to strengthen the reference attack during the iterations of ILA, by interpolating the reference attack itself with a stronger attack. The output of the previous iteration, $\mathbf{x}''$, is a good candidate for a strong attack. At iteration $t$, we set

$$\mathbf{x}'_{t+1} = \lambda \mathbf{x}''_t + (1 - \lambda)\mathbf{x}'_t \tag{8}$$

where $\lambda$ is a weighting factor in the range $[0, 1]$ to control the proportion of the two attacks. This augmentation is similar to mixup (Zhang et al., 2018), except that we are mixing two adversarial examples, rather than two images from different classes. With the assumption that the attack is fine-tuned to be more transferable in every iteration, by interpolating two attacks, we can strengthen the transferability of the reference attack while preserving a part of its original characteristics.

Based on our preliminary findings, setting $\lambda$ to non-trivial values such as $0.5$ suffices to yield satisfactory performance. However, a constant value lacks consideration of the behavior of the two attacks. Precisely, we hope to perform an adaptive interpolation depending on the effectiveness of $\mathbf{x}''_t$. If $\mathbf{x}''_t$ is weak after a single update, $\lambda$ should bias towards preserving the reference attack more than mixing with the new attack. Since ILA focuses on maximizing the linear projection between feature map discrepancies, the norm of the feature map discrepancy can be a good indicator of the performance. Consequently, we set

$$\lambda = \frac{\|\Delta\mathbf{y}''_l\|_2}{\|\Delta\mathbf{y}''_l\|_2 + \|\Delta\mathbf{y}'_l\|_2} \tag{9}$$

If we neglect the transformation in Section 3.2 for simplicity, $\Delta\mathbf{y}'_l$ and $\Delta\mathbf{y}''_l$ will remain the same as what we obtained from Equations (2) and (3), respectively. Note that the value of $\lambda$ is recomputed in each iteration before interpolation is applied.

A similar approach of reusing previous attacks in the temporal domain can be applied to the reverse adversarial update in Section 3.2. The reference attack $\mathbf{x}'$ in Equation (7) can be further extended to a collection of $\mathbf{x}'_t$ for every iteration $t$. This is not the only work to consider the temporary values of the reference attack before its convergence. ILA++ (Li et al., 2020b) also collects all losses computed during the generation of the reference attack, achieving superiority over the standard ILA. Different from Li et al. (2020b) which uses past losses (with respect to $\mathbf{x}'$) to help update $\mathbf{x}''_t$, we make use of the past $\mathbf{x}'_t$ directly as an augmentation to enrich the information regarding feature discrepancies under different attacks.

## 4 EXPERIMENTAL RESULTS

In this section, we evaluate the attack transferability of ILA-DA in comparison with previous methods based on fine-tuning as well as state-of-the-art transfer-based attack generation methods on multiple defended and undefended models. Next, we investigate the effects of the three augmentation techniques and identify influential factors for the transferability of an attack.

### 4.1 SETUP

**Undefended models.** Following the line of work in transfer-based attacks, a total of nine classification models are used to evaluate attack transferability, including ResNet50 (He et al., 2016),

VGG19 (Simonyan & Zisserman, 2015), Inception V3 (Inc-v3) (Szegedy et al., 2016), Wide ResNet50 2x (WRN) (Zagoruyko & Komodakis, 2016), DenseNet161 (DenseNet) (Huang et al., 2017), ResNeXt101 (ResNeXt) (Xie et al., 2017), MobileNet V2 (MobileNet) (Sandler et al., 2018), PNASNet5 (PNASNet) (Liu et al., 2018) and SENet50 (SENet) (Hu et al., 2018). Among these models, we use ResNet50, VGG19 and Inception V3 as the source models of the attack. All the models are pretrained on ImageNet (Russakovsky et al., 2015), with the model parameters of PNASNet[1] and SENet[2] obtained from public repositories and the remaining from Torchvision (Paszke et al., 2019). For the choice of the intermediate layer, we opt the layer 3-1 for ResNet50, layer 9 for VGG19, and layer 6a for Inception V3, where the former two have been shown to result in good performance by Li et al. (2020b). Due to the ineffectiveness of transfer-based targeted attacks (Liu et al., 2017), we report the untargeted attacks with $\epsilon$ values of $8/255$ (0.03) under the $\ell_\infty$ norm constraint. To measure the attack success rate, we randomly sample 5000 images from the ILSVRC2012 validation set with all images being classified correctly by the nine models. The default number of iterations of I-FGSM is 10 and the attack step size is set to $\max(\frac{1}{255}, \frac{\epsilon}{\text{no. of iterations}})$. To mount a complete attack, we first run I-FGSM for 10 iterations on the source model, and then pass the example as the reference attack to ILA-DA to perform fine-tuning for 50 iterations. The same model is used as both the source model of I-FGSM and surrogate model of ILA-DA.

To demonstrate the superiority of ILA-DA over the ILA family, we include the result of ILA (Huang et al., 2019) and ILA++ (Li et al., 2020b). Besides the ILA family, we also append the results of other state-of-the-art transfer-based attacks, namely MI-CT-FGSM and NI-CT-FGSM, with their corresponding variations with variance tuning, SGM and LinBP as introduced in Section 2.1. Since the experimental setting in the CTM family is slightly different from ours and LinBP, we reproduce the works in the CTM family and apply in our experimental environment. The results of the attacks on undefended models are shown in Table 1, with the details of the hyper-parameters listed in Appendix F. We also conduct extensive experiments using other perturbation budgets, source models, target models and datasets. The results can be found in Appendix D, H, K and J respectively.

**Defended models.** We follow the experimental setup used in the previous works (Dong et al., 2018; 2019; Wang & He, 2021), which includes 1000 pre-sampled test images and a set of models which are pre-trained under ensemble adversarial training (Tramer et al., 2018). Then we compare ILA-DA with the baselines introduced in Section 2.1, including I-FGSM, MI-FGSM, MI-CT-FGSM, NI-CT-FGSM, Admix and variance tuning. We pick Inception V3 as the source model and an adversarially trained Inception V3 as the surrogate model of ILA-DA, and all attacks are performed with perturbation size $\epsilon = 16/255 (\approx 0.063)$. The attacks are evaluated against three robust models: Inc-v3$_{ens3}$, Inc-v3$_{ens4}$, and IncRes-v2$_{ens}$. All three models undergo ensemble adversarial training on ImageNet. Besides, we also test the attacks against the top three submitted defenses in the NIPS 2017 adversarial competition: HGD (Liao et al., 2018), R&P (Xie et al., 2018) and MMD[3] with the default models and hyper-parameters used in their corresponding repositories. Since the gradients of the adversarially trained models are more difficult to exploit, we increase the number of ILA-DA iterations to 500 in order to guarantee convergence of the attack. The attack success rate against each of the defenses is reported in Table 2, while a study of the number of iterations is reported in Appendix A.

## 4.2 BLACK-BOX TRANSFERABILITY OF ILA-DA ON UNDEFENDED MODELS

ILA-DA outperforms ILA and ILA++ on 8 transferred models by a large margin. Although the adversarial examples generated by ILA-DA are slightly less effective on the source model (ResNet50), the examples achieve remarkable attack success rates on other transferred models (see Table 1). In particular, ILA-DA attains an average attack success rate of 84.5%, which is 19.5% better than ILA and 4.7% better than the previous state-of-the-art (VNI-CT-FGSM). The improved attack transferability validates the effectiveness of our proposed augmentation techniques. It is worth mentioning that the attack difficulty increases with the differences in model architectures. For example, the attacks generated from ResNet (with skip connections) tend to transfer better to models also with skip connections, such as WRN, DenseNet, ResNeXt, compared to Inception V3 and PNASNet. Even under such model dissimilarity, the improvement in attack transferability caused by ILA-DA is more

---

[1] https://github.com/Cadene/pretrained-models.pytorch
[2] https://github.com/moskomule/senet.pytorch
[3] https://github.com/anlthms/nips-2017/tree/master/mmd

Table 1: Attack success rates of ImageNet adversarial examples on nine undefended models, generated from ResNet50 with $\epsilon = 8/255$ (0.03). The column 'Average' is the average of all models except the source model.

| Method | ResNet50* | Inc-v3 | WRN | VGG19 | PNASNet |
|---|---|---|---|---|---|
| I-FGSM | 99.9% | 14.9% | 41.3% | 26.4% | 17.4% |
| I-FGSM + ILA | 99.9% | 34.6% | 79.9% | 66.7% | 43.3% |
| I-FGSM + ILA++ | 99.9% | 41.5% | 87.1% | 75.2% | 49.2% |
| I-FGSM + LinBP + SGM | **100.0%** | 35.3% | 88.7% | 78.7% | 45.0% |
| MI-CT-FGSM | 97.9% | 65.0% | 77.5% | 69.7% | 67.6% |
| NI-CT-FGSM | 99.4% | 59.6% | 78.1% | 67.1% | 64.9% |
| VMI-CT-FGSM | 99.4% | 66.6% | 84.8% | 73.0% | 70.2% |
| VNI-CT-FGSM | 99.8% | **67.3%** | 87.4% | 76.1% | 71.8% |
| I-FGSM + ILA-DA (Ours) | 99.0% | 64.1% | **92.6%** | **91.6%** | **72.4%** |

| Method (cont.) | DenseNet | ResNeXt | MobileNet | SENet | Average |
|---|---|---|---|---|---|
| I-FGSM | 31.4% | 41.8% | 31.9% | 44.0% | 31.1% |
| I-FGSM + ILA | 69.2% | 78.5% | 67.6% | 80.2% | 65.0% |
| I-FGSM + ILA++ | 79.1% | 78.5% | 75.3% | 87.6% | 71.7% |
| I-FGSM + LinBP + SGM | 81.4% | 77.1% | 75.1% | 91.0% | 74.7% |
| MI-CT-FGSM | 77.2% | 71.6% | 74.0% | 80.8% | 72.9% |
| NI-CT-FGSM | 76.0% | 67.0% | 73.6% | 81.1% | 70.9% |
| VMI-CT-FGSM | 82.9% | 77.3% | 80.5% | 86.7% | 77.8% |
| VNI-CT-FGSM | 85.1% | 80.3% | 81.5% | 88.9% | 79.8% |
| I-FGSM + ILA-DA (Ours) | **89.8%** | **86.2%** | **87.9%** | **91.3%** | **84.5%** |

\* The source model used to generate the attack.

Table 2: Attack success rates of ImageNet adversarial examples on six defended methods, generated from Inception V3 with $\epsilon = 16/255$ (0.063).

| Attack | Inc-v3$_{ens3}$ | Inc-v3$_{ens4}$ | IncRes-v2$_{ens}$ | HGD | R&P | MMD | Average |
|---|---|---|---|---|---|---|---|
| I-FGSM | 12.1% | 10.9% | 5.8% | 2.7% | 4.0% | 8.3% | 7.3% |
| + ILA-DA | **86.3%** | **81.8%** | **66.4%** | **82.2%** | **68.3%** | **70.9%** | **75.9%** |
| MI-FGSM | 14.1% | 13.0% | 6.6% | 4.6% | 5.0% | 8.3% | 8.6% |
| + ILA-DA | **83.6%** | **79.2%** | **64.6%** | **79.5%** | **65.9%** | **70.2%** | **73.8%** |
| MI-CT-FGSM | 65.5% | 62.1% | 45.5% | 56.6% | 44.5% | 52.5% | 54.5% |
| + ILA-DA | **88.1%** | **84.4%** | **72.3%** | **84.4%** | **73.3%** | **76.2%** | **79.8%** |
| NI-CT-FGSM | 58.8% | 54.4% | 40.0% | 49.2% | 38.0% | 46.1% | 47.8% |
| + ILA-DA | **87.3%** | **83.9%** | **68.5%** | **81.0%** | **71.0%** | **74.7%** | **77.7%** |
| MI-Admix-TI-DIM | 73.4% | 70.7% | 53.9% | 65.4% | 53.7% | 58.0% | 62.5% |
| + ILA-DA | **88.9%** | **86.3%** | **74.6%** | **85.2%** | **77.1%** | **79.7%** | **81.9%** |
| VMI-CT-FGSM | 77.6% | 75.2% | 63.6% | 72.1% | 63.0% | 69.7% | 70.2% |
| + ILA-DA | **89.1%** | **85.7%** | **74.5%** | **84.1%** | **75.6%** | **78.7%** | **81.3%** |
| VNI-CT-FGSM | 79.1% | 77.4% | 65.3% | 72.7% | 63.5% | 70.8% | 71.5% |
| + ILA-DA | **88.0%** | **86.1%** | **74.5%** | **84.2%** | **75.8%** | **78.0%** | **81.1%** |

prominent than the ILA baselines. We believe such a phenomenon can be attributed to the extensive augmentations that enable both state-of-the-art attacks and ILA-DA to exploit gradients with better resemblance to different architectures.

## 4.3 Black-box Transferability of ILA-DA on Defended Models

In Table 2, ILA-DA also shows outstanding attack transferability against six defended models. Unlike the CTM family, ILA-DA does not require stacking multiple attack methods to achieve high attack transferability. Based on a weak I-FGSM reference attack, I-FGSM + ILA-DA yields an average attack success rate of 75.9%, which outperforms the strongest state-of-the-art (VNI-CT-FGSM) by 4.4%. Moreover, incorporating ILA-DA to existing methods brings further improvements. Specifically, using the attack reference generated by MI-Admix-TI-DIM, ILA-DA can attain the highest attack success rate of 81.9%.

Table 3: Comparison of the attack success rates of ImageNet adversarial examples on various models using ILA-DA with different augmentation configurations.

| Method | ResNet50* | Inc-v3 | WRN | VGG19 | PNASNet |
|---|---|---|---|---|---|
| I-FGSM + ILA-DA | 99.0% | **64.1%** | **92.6%** | 91.6% | **72.4%** |
| w/o Augmentation | 99.7% | 39.7% | 86.6% | 79.8% | 49.7% |
| w/ Random Augmentation | 98.4% | 62.6% | 90.4% | 90.9% | 71.3% |
| w/ All Augmentation | 93.4% | 59.4% | 78.0% | 79.9% | 64.3% |
| w/o Reverse adversarial update | 97.3% | 50.6% | 88.0% | **92.1%** | 65.9% |
| w/o Attack interpolation | **99.9%** | 54.0% | 91.8% | 83.8% | 61.6% |

| Method (cont.) | DenseNet | ResNeXt | MobileNet | SENet | Average |
|---|---|---|---|---|---|
| I-FGSM + ILA-DA | **89.8%** | **86.2%** | 87.9% | 91.3% | **84.5%** |
| w/o Augmentation | 77.0% | 76.0% | 75.6% | 85.9% | 71.3% |
| w/ Random Augmentation | 87.3% | 83.8% | 88.0% | 89.3% | 82.9% |
| w/ All Augmentation | 74.3% | 70.2% | 78.0% | 78.7% | 72.8% |
| w/o Reverse adversarial update | 85.7% | 82.1% | **88.3%** | 88.6% | 79.9% |
| w/o Attack interpolation | 87.2% | 86.0% | 83.9% | **91.8%** | 80.0% |

*  The source model used to generate the attack.

As the defended models are trained to be robust to adversarial inputs, more ILA-DA fine-tuning is required in comparison to undefended models (see Appendix A and Appendix N). Although using a larger number of ILA-DA iterations induces higher computational overhead, ILA-DA could still achieve comparable performance with state-of-the-art attacks such as VNI-CT-FGSM and LinBP under similar running time, while ILA-DA also exhibits a great degree of improvement when the time budget is up-scaled.

## 4.4   ABLATION STUDY

ILA-DA can be regarded as an extension of the original ILA, equipped with three ILA augmentation techniques: automated data augmentation, reverse adversarial update, and attack interpolation. We conduct an ablation study on the effects of the three augmentations by removing each of them from ILA-DA and reporting the performance of the reduced methods in Table 3. In addition, we also study two non-automated ways of applying translation, cropping and rotation to the ILA references. In particular, Random Augmentation randomly applies one of the three augmentations; All Augmentation applies all three augmentations to the reference images.

While all the three proposed augmentation techniques contribute to the improved attack transferability on average, Table 3 reveals that applying data augmentation to the ILA references increases the attack success rate the most, followed by the use of attack interpolation and reverse adversarial update. Although reverse adversarial update and attack interpolation slightly lower the performance by 0.5% in 2 out of 16 settings, the improvements in transferability across different models brought by the two augmentations are evident. It is shown that applying all augmentations may cause excessive distortion to the images and lead to a significant drop in the attack success rate. Nevertheless, using the learned augmentation parameter shows non-trivial improvements over random augmentation, validating the use of the proposed automated data augmentation scheme.

## 5   CONCLUSION

In this paper, we presented ILA-DA, which applies three types of data augmentation techniques to the original ILA formulation. Specifically, we proposed an automated data augmentation method which is searched by learnable parameters, reverse adversarial update which exploits the adversarial noise to augment the reference, and attack interpolation which strengthens the reference attack across iterations. Incorporating all the proposed augmentations, ILA-DA outperforms state-of-the-art methods for transfer-based attacks on both undefended and defended models. We also studied the contribution of each augmentation technique to ILA-DA attack transferability. This work showcases how data augmentation techniques could benefit the generation of transfer-based adversarial attacks. We believe that it is beneficial to extend the proposed automated method to search for more types of data augmentation, other augmentation parameters, and even the optimal ILA layer to generate the attack.

ACKNOWLEDGMENTS

This research has been made possible by funding support from the Research Grants Council of Hong Kong under the General Research Fund project 16204720 and the Research Impact Fund project R6003-21.

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

## A    NUMBER OF ILA ITERATIONS

With extensive augmentation, we also search for the number of iterations for ILA-DA to result in a stable attack. We vary the number of iterations from 5 to 2000, in both the setup of undefended ($\epsilon = 8/255$) and defended models ($\epsilon = 16/255$). A plot of the result is shown in Figure 2. For undefended models, ILA-DA could yield attacks with high transferability within 50 iterations. However, for defended models, it requires more iterations to converge to an effective attack.

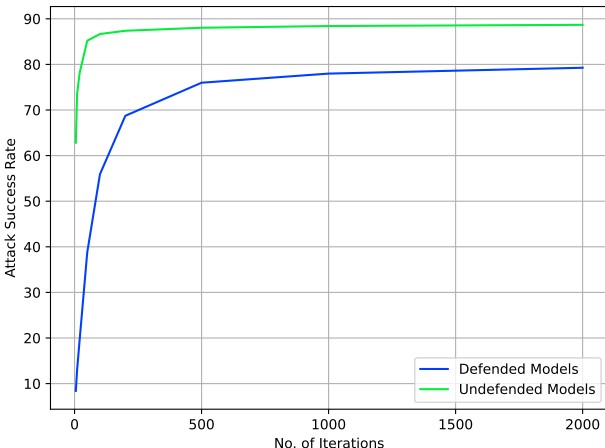

Figure 2: Average attack success rates for the defended and undefended models over different number of ILA iterations.

## B    EFFECT OF THE CHOICE OF INTERMEDIATE LAYER

We test ILA and ILA-DA on several neighbouring layers of the default layer suggested by Huang et al. (2019). As shown in Figure 3, ILA-DA is strictly better than ILA in terms of attack success rate in all tested layers. Furthermore, the inverted U-curve patterns exhibited in Figure 3 suggest that the method is not very sensitive to the selection of layer near the layer proposed by ILA.

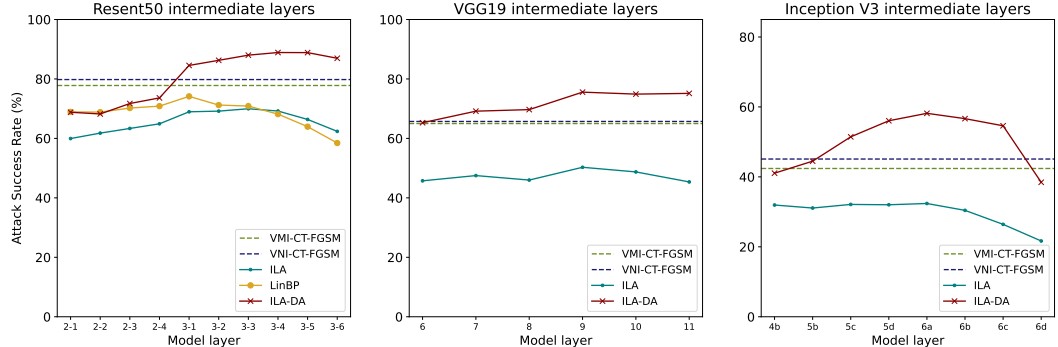

Figure 3: Attack transferability over different choices of the intermediate layer for ResNet50, VGG19 and Inception V3. The dotted curve shows the performance of ILA without augmentation. '3-1' in ResNet50 refers to the first residual block of the third meta-block. The naming of layers of the models follows the PyTorch convention.

## C    ANALYSIS OF THE LEARNED AUGMENTATION POLICY

ILA-DA optimizes three image transformation operations that are commonly used, including translation, cropping and rotation. On the ImageNet dataset, we discovered that the cropping operation yields the strongest attack. Empirical results also show that the augmentation parameters converged to certain fixed points. In Figure 4a, ILA-DA explores different augmentations in the first few iterations and discovers random cropping as the best transformations for maximizing the ILA objective. Figure 4b further shows that the converged augmentation parameters are applied to the attack references throughout the ILA-DA fine-tuning.

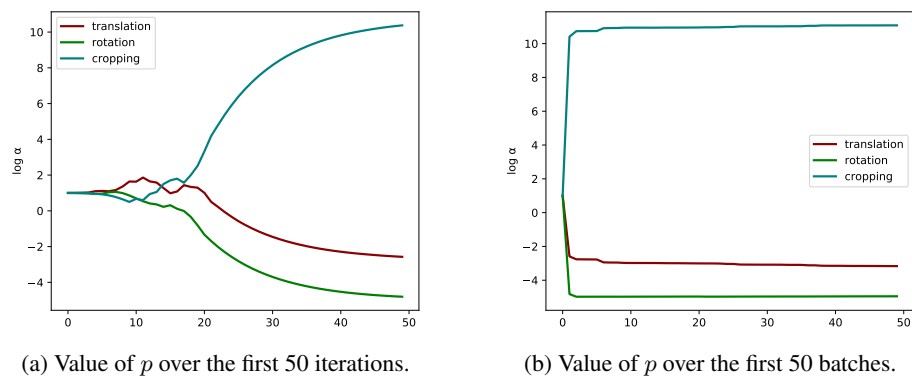

(a) Value of $p$ over the first 50 iterations.  (b) Value of $p$ over the first 50 batches.

Figure 4: Convergence of the learned augmentation parameters.

## D    RESULTS ON UNDEFENDED MODELS WITH OTHER PERTURBATION SIZES

As a supplementary to Table 1 which shows the attack results with $\epsilon = 8/255$ (0.03) under $\ell_\infty$ norm constraint on undefended models, we also experiment with different perturbation budgets and norm constraint. Table 4 compares the performance of ILA-DA with other baselines with $\epsilon = 13/255$ (0.05) under $\ell_\infty$ norm constraint, while Table 5 performs the same comparison but with $\epsilon = 5.0$ under $\ell_2$ norm constraint. Both experimental results show that ILA-DA leads I-FGSM, ILA, ILA++ and VMI-CT-FGSM by a large margin.

Table 4: Attack success rates of ImageNet adversarial examples on nine undefended models, generated from ResNet50 with $\epsilon = 13/255$ (0.05). The column 'Average' is the average of all models except the source model.

| Method | ResNet50* | Inc-v3 | WRN | VGG19 | PNASNet |
|---|---|---|---|---|---|
| I-FGSM | **100.0%** | 25.8% | 72.1% | 48.3% | 29.6% |
| I-FGSM + ILA | **100.0%** | 51.5% | 93.2% | 86.6% | 65.2% |
| I-FGSM + ILA++ | 99.9% | 63.6% | 96.3% | 90.5% | 70.3% |
| I-FGSM + LinBP + SGM | **100.0%** | 63.4% | 98.5% | 94.7% | 73.0% |
| MI-CT-FGSM | 99.7% | 81.3% | 91.0% | 85.7% | 83.6% |
| NI-CT-FGSM | **100.0%** | 80.9% | 92.9% | 89.4% | 82.0% |
| VMI-CT-FGSM | **100.0%** | 84.7% | 94.4% | 88.0% | 85.0% |
| VNI-CT-FGSM | **100.0%** | 85.7% | 95.1% | 89.7% | 87.2% |
| I-FGSM + ILA-DA (Ours) | 99.7% | **88.4%** | **98.7%** | **98.6%** | **91.6%** |

| Method (cont.) | DenseNet | ResNeXt | MobileNet | SENet | Average |
|---|---|---|---|---|---|
| I-FGSM | 57.8% | 70.9% | 54.5% | 71.9% | 53.8% |
| I-FGSM + ILA | 85.5% | 91.0% | 84.0% | 91.9% | 81.1% |
| I-FGSM + ILA++ | 92.3% | 92.3% | 89.3% | 96.2% | 86.3% |
| I-FGSM + LinBP + SGM | 96.6% | 94.9% | 93.2% | **98.9%** | 89.1% |
| MI-CT-FGSM | 91.1% | 86.4% | 87.5% | 93.5% | 87.5% |
| NI-CT-FGSM | 93.1% | 86.4% | 90.5% | 95.4% | 88.8% |
| VMI-CT-FGSM | 93.6% | 90.9% | 91.7% | 95.3% | 90.5% |
| VNI-CT-FGSM | 95.4% | 93.1% | 92.0% | 96.5% | 91.8% |
| I-FGSM + ILA-DA (Ours) | **98.1%** | **96.9%** | **97.7%** | 98.4% | **96.0%** |

* The source model used to generate the attack.

Table 5: Attack success rates of ImageNet adversarial examples on nine undefended models, generated from ResNet50 with $\epsilon = 5.0$ under $\ell_2$ norm constraint. The column 'Average' is the average of all models except the source model.

| Method | ResNet50* | Inc-v3 | WRN | VGG19 | PNASNet |
|---|---|---|---|---|---|
| I-FGSM | **100.0%** | 15.0% | 38.4% | 26.1% | 17.1% |
| I-FGSM + ILA | 99.9% | 17.9% | 57.7% | 42.2% | 23.6% |
| I-FGSM + ILA++ | **100.0%** | 17.1% | 61.2% | 43.7% | 21.1% |
| I-FGSM + LinBP | **100.0%** | 14.0% | 49.3% | 37.5% | 18.7% |
| MI-CT-FGSM | 92.7% | **43.1%** | 55.0% | 47.5% | **47.8%** |
| NI-CT-FGSM | 93.9% | 34.7% | 41.2% | 39.8% | 42.6% |
| VMI-CT-FGSM | 95.4% | 42.5% | 52.4% | 46.0% | 46.2% |
| VNI-CT-FGSM | 96.9% | 37.8% | 47.4% | 42.0% | 43.4% |
| I-FGSM + ILA-DA (Ours) | 96.3% | 34.0% | **69.6%** | **68.4%** | 42.2% |

| Method (cont.) | DenseNet | ResNeXt | MobileNet | SENet | Average |
|---|---|---|---|---|---|
| I-FGSM | 29.5% | 27.7% | 30.6% | 41.5% | 36.2% |
| I-FGSM + ILA | 44.2% | 45.2% | 44.4% | 59.3% | 48.3% |
| I-FGSM + ILA++ | 46.6% | 48.2% | 45.8% | 62.3% | 49.5% |
| I-FGSM + LinBP | 36.7% | 34.7% | 33.8% | 52.3% | 41.9% |
| MI-CT-FGSM | 56.4% | 45.5% | 53.5% | 61.7% | 55.9% |
| NI-CT-FGSM | 41.6% | 31.4% | 43.2% | 49.2% | 46.4% |
| VMI-CT-FGSM | 52.9% | 42.0% | 51.3% | 59.2% | 54.2% |
| VNI-CT-FGSM | 47.0% | 35.2% | 47.4% | 52.7% | 50.0% |
| I-FGSM + ILA-DA (Ours) | **62.4%** | **58.8%** | **61.8%** | **69.7%** | **62.6%** |

* The source model used to generate the attack.

# E  EFFECT OF REVERSE ADVERSARIAL UPDATE ON THE MODEL PERFORMANCE

We would like to verify the intuition of using reverse adversarial update, specifically, whether in practice such operation can decrease the loss and increase the model confidence. Therefore, we randomly sample 5000 images, including those that are misclassified by the model. Then we apply I-FGSM10 in the form of reverse adversarial update to the images, and pass them back to the model for classification. We record the loss and apply softmax to the logits to obtain the confidence, and report the average result of all 5000 images. The result is summarized in Table 6.

Furthermore, we also inspect the class activation map (CAM) of the source model. Figure 5 visualizes some of the results. For some images (the first two), updating reversely helps the model focus on more features. For the remaining (the latter two), adversarial reverse update does not exhibit a significant change of intermediate layers' contribution to the ground-truth class.

Table 6: Changes in loss and confidence after applying reverse adversarial update on the images for ResNet50.

| Model | Image | Loss | Confidence | Accuracy |
|---|---|---|---|---|
| ResNet50 | Clean | 0.9755 | 0.7913 | 75.08% |
| | Reversely updated | 0.3070 | 0.8847 | 92.20% |
| Inception V3 | Clean | 1.1165 | 0.7339 | 76.44% |
| | Reversely updated | 0.3525 | 0.8296 | 94.60% |
| VGG19 | Clean | 1.1431 | 0.7419 | 70.54% |
| | Reversely updated | 0.5756 | 0.8998 | 89.98% |

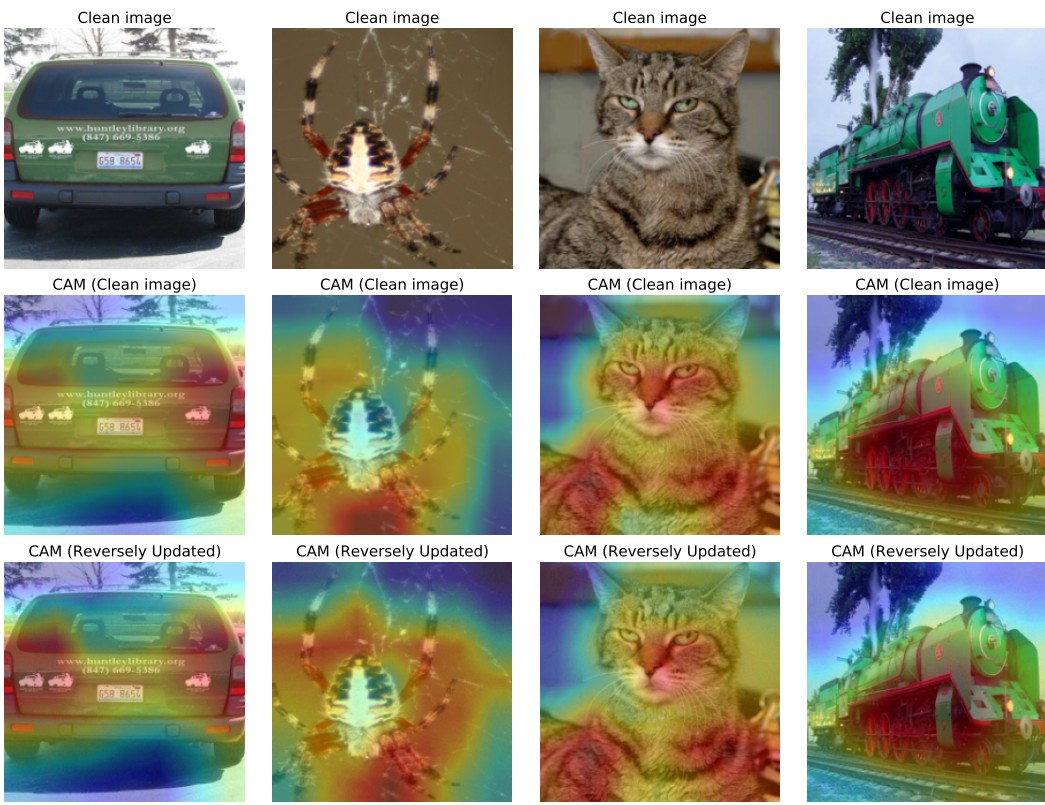

Figure 5: CAM visualization of the images after reverse adversarial update.

To test the effect of reversely updated examples on targeted models, we also feed them into the remaining undefended models and observe the change in accuracy. The result is reported in Table 7, which shows that reverse adversarial update not only benefits the source model, but also many of the other models with similar architecture.

Table 7: Top-1 accuracy of the models under the example with reverse adversarial update from different source models. The source model "-" indicates clean example without any perturbation.

| Source model | Target model | | | | |
| | ResNet50 | Inception v3 | WRN | VGG19 | PNASNet |
| --- | --- | --- | --- | --- | --- |
| - | 75.08% | 76.44% | 77.58% | 70.54% | 71.34% |
| ResNet50 | 92.20% | 79.48% | 82.22% | 74.06% | 72.84% |
| Inception V3 | 78.58% | 94.60% | 80.70% | 73.62% | 72.88% |
| VGG19 | 78.46% | 79.56% | 80.06% | 89.98% | 71.76% |

| Source model | Target model | | | | |
| | DenseNet | ResNeXt | MobileNet | SENet | Average |
| --- | --- | --- | --- | --- | --- |
| - | 75.86% | 78.30% | 70.48% | 75.36% | 74.55% |
| ResNet50 | 80.70% | 82.70% | 73.70% | 80.70% | 79.84% |
| Inception V3 | 80.54% | 81.54% | 73.04% | 78.80% | 79.37% |
| VGG19 | 80.58% | 81.02% | 74.38% | 79.00% | 79.42% |

## F  HYPER-PARAMETERS OF THE BASELINES

We reproduced the algorithms of CTM (DIM, TIM, SIM), MI-FGSM, NI-FGSM and variance tuning according to the repository released by Wang & He (2021), and tested the attacks under our experimental settings. The hyper-parameters used in the experiments are shown in Table 8, which are the default values in their corresponding papers. For the implementation of SGM and LinBP, we directly use the official implementation released by Guo et al. (2020).

Table 8: Hyper-parameters used in the baselines.

| Method | Hyper-parameter | Value |
| --- | --- | --- |
| MI-FGSM | $\alpha$ | 1.0 |
| NI-FGSM | $\alpha$ | 1.0 |
| DIM | Probability | 0.5 |
| | Upscale ratio | 1.1 |
| TIM | Kernel size | $7 \times 7$ |
| SIM | Scale copies | $5\ (i = 0, 1, 2, 3, 4)$ |
| Variance Tuning | $N$ | 20 |
| | $\beta$ | 1.5 |
| Admix | $\eta$ | 0.2 |
| SGM | $\lambda$ | 0.5 |
| LinBP | I-FGSM iteration | 300, 100 (ImageNet, CIFAR-10) |
| | Layer | '3-1' for ResNet50 |

## G  APPLYING AUTOMATED DATA AUGMENTATION TO OTHER BASELINES

Through extensive experiments, we show the effectiveness of using Automated Data Augmentation to strengthen ILA attacks. In this section, we study whether the proposed Automated Data Augmentation method can be extended to other adversarial attack frameworks beyond ILA attacks.

Specifically, we compare the attack transferability of multiple baseline attacks with and without the automated augmentation policy (dubbed AutoDA). The result is shown in Table 9.

From Table 9, we observe a significant improvement in attack transferability when AutoDA is applied to attacks that have no data augmentation, like I-FGSM and MI-FGSM. However, for some attacks that already employed a fixed augmentation scheme, like the CTM family, AutoDA could not improve the performance. We suspect that using multiple augmentations may cause excessive distortions to the attacks, thus lowering the attack success rate. That said, ILA-DA remains the strongest method among all the tested combinations. This can be attributed to the use of ILA projection loss that further perturbs the image feature, as well as the use of reverse update and attack interpolation that create stronger attacks based on the reference attacks.

Table 9: Attack success rates of ImageNet adversarial examples on undefended models, generated from VGG19 with $\epsilon = 8/255$ (0.03).

| Method | ResNet50* | Inc-v3 | WRN | VGG19 | PNASNet | DenseNet | ResNeXt | MobileNet | SENet | Avg. |
|---|---|---|---|---|---|---|---|---|---|---|
| I-FGSM | 99.9% | 14.9% | 41.3% | 26.4% | 17.4% | 31.4% | 41.8% | 31.9% | 44.0% | 31.1% |
| + AutoDA | 99.8% | 45.7% | 77.3% | 67.5% | 51.5% | 72.6% | 68.1% | 70.9% | 77.3% | 66.4% |
| MI-FGSM | 100.0% | 43.2% | 73.8% | 60.6% | 51.7% | 67.5% | 65.0% | 64.7% | 76.6% | 62.9% |
| + AutoDA | 100.0% | 64.4% | 86.6% | 82.1% | 67.6% | 84.5% | 80.5% | 84.2% | 85.3% | 79.4% |
| NI-FGSM | 100.0% | 43.5% | 75.0% | 62.6% | 52.3% | 67.0% | 64.9% | 66.4% | 76.3% | 63.5% |
| + AutoDA | 99.8% | 57.3% | 78.3% | 77.2% | 65.7% | 77.6% | 72.1% | 77.8% | 79.7% | 73.2% |
| VMI-CT-FGSM | 99.4% | 66.6% | 84.8% | 73.0% | 70.2% | 82.9% | 77.3% | 80.5% | 86.7% | 77.8% |
| + AutoDA | 99.2% | 67.2% | 83.9% | 73.2% | 69.5% | 82.4% | 77.9% | 78.9% | 86.3% | 77.4% |
| VNI-CT-FGSM | 99.8% | 67.3% | 87.4% | 76.1% | 71.8% | 85.1% | 80.3% | 81.5% | 88.9% | 79.8% |
| + AutoDA | 85.1% | 55.8% | 70.2% | 61.1% | 60.8% | 68.2% | 63.5% | 65.6% | 71.4% | 64.6% |
| ILA-DA | 99.0% | 64.1% | 92.6% | 91.6% | 72.4% | 89.8% | 86.2% | 87.9% | 91.3% | 84.5% |

\* The source model used to generate the attack.

# H    ATTACKS USING OTHER SOURCE MODELS

In this section, we test the effectiveness of ILA-DA with other source models. In particular, we adopt VGG19 and Inception V3 as the source models for both the reference attack and ILA-DA, and re-evaluate the attack success rate on the remaining eight undefended models. The results are shown in Table 10 and Table 11. For Inception V3, ILA-DA still outperforms every other baseline. For VGG19, the performance of I-FGSM + ILA-DA is slightly worse than that of the best attack, VNI-CT-FGSM. However, when we fine-tune ILA-DA on a stronger reference attack such as MI-CT-FGSM, our method regains the best average attack success rate among all other methods.

Table 10: Attack success rates of ImageNet adversarial examples on undefended models, generated from VGG19 with $\epsilon = 8/255$ (0.03).

| Method | ResNet50 | Inc-v3 | WRN | VGG19[*] | PNASNet |
|---|---|---|---|---|---|
| I-FGSM + ILA | 53.8% | 25.9% | 49.5% | 99.3% | 46.3% |
| I-FGSM + ILA++ | 57.9% | 27.9% | 52.9% | 99.3% | 49.5% |
| MI-CT-FGSM | 64.4% | 59.5% | 55.6% | 99.3% | 68.7% |
| NI-CT-FGSM | 60.7% | 53.9% | 49.7% | **100.0%** | 66.7% |
| VMI-CT-FGSM | 68.9% | 62.0% | 60.5% | 99.7% | 71.2% |
| VNI-CT-FGSM | 70.3% | **62.3%** | 60.7% | **100.0%** | **71.9%** |
| I-FGSM + ILA-DA (Ours) | 68.1% | 32.0% | 67.5% | 98.2% | 59.0% |
| MI-CT-FGSM + ILA-DA (Ours) | **72.8%** | 48.5% | **69.1%** | 99.6% | 69.6% |

| Method (cont.) | DenseNet | ResNeXt | MobileNet | SENet | Average |
|---|---|---|---|---|---|
| I-FGSM + ILA | 40.1% | 36.3% | 66.5% | 49.4% | 46.0% |
| I-FGSM + ILA++ | 44.9% | 39.7% | 68.7% | 53.6% | 49.4% |
| MI-CT-FGSM | 58.7% | 42.9% | 74.2% | 63.2% | 60.9% |
| NI-CT-FGSM | 51.1% | 38.4% | 70.2% | 57.1% | 56.0% |
| VMI-CT-FGSM | 62.2% | 48.4% | 78.2% | 68.6% | 65.0% |
| VNI-CT-FGSM | 62.5% | 48.9% | 79.3% | 69.4% | 65.7% |
| I-FGSM + ILA-DA (Ours) | 58.6% | 57.9% | 81.9% | 66.7% | 61.5% |
| MI-CT-FGSM + ILA-DA (Ours) | **65.3%** | **58.5%** | **82.1%** | **71.7%** | **67.2%** |

[*] The source model used to generate the attack.

Table 11: Attack success rates of ImageNet adversarial examples on undefended models, generated from Inception V3 with $\epsilon = 8/255$ (0.03).

| Method | ResNet50 | Inc-v3[*] | WRN | VGG19 | PNASNet |
|---|---|---|---|---|---|
| I-FGSM + ILA | 34.1% | 96.9% | 29.6% | 37.5% | 34.0% |
| I-FGSM + ILA++ | 36.7% | 97.1% | 31.9% | 38.9% | 35.1% |
| MI-CT-FGSM | 36.4% | 94.2% | 34.5% | 41.9% | 51.3% |
| NI-CT-FGSM | 33.9% | 98.2% | 30.5% | 38.6% | 50.7% |
| VMI-CT-FGSM | 39.6% | 96.7% | 37.7% | 43.8% | 52.6% |
| VNI-CT-FGSM | 42.6% | **98.8%** | 40.6% | 46.6% | **56.4%** |
| I-FGSM + ILA-DA (Ours) | **58.4%** | 91.1% | **55.8%** | **70.1%** | 53.1% |

| Method (cont.) | DenseNet | ResNeXt | MobileNet | SENet | Average |
|---|---|---|---|---|---|
| I-FGSM + ILA | 27.0% | 23.7% | 43.3% | 29.6% | 32.4% |
| I-FGSM + ILA++ | 29.1% | 25.6% | 43.8% | 31.5% | 34.1% |
| MI-CT-FGSM | 38.7% | 30.2% | 43.8% | 37.5% | 39.3% |
| NI-CT-FGSM | 33.9% | 25.4% | 42.1% | 34.0% | 36.1% |
| VMI-CT-FGSM | 43.3% | 35.2% | 46.2% | 40.5% | 42.4% |
| VNI-CT-FGSM | 45.7% | 36.1% | 49.0% | 43.4% | 45.1% |
| I-FGSM + ILA-DA (Ours) | **49.3%** | **42.8%** | **71.4%** | **51.1%** | **56.5%** |

[*] The source model used to generate the attack.

## I   NAME MATCHING OF THE CTM FAMILY

Table 12: A table matching the names between the baseline attacks and the previous works.

| Methods | Momentum | Nesterov Accelerated | DIM | TIM | SIM | Variance Tuning | Admix |
|---|---|---|---|---|---|---|---|
| Source | Dong et al. (2018) | Lin et al. (2020) | Xie et al. (2019a) | Dong et al. (2019) | Lin et al. (2020) | Wang & He (2021) | Wang et al. (2021) |
| I-FGSM | × | × | × | × | × | × | × |
| MI-FGSM | ✓ | × | × | × | × | × | × |
| MI-CT-FGSM | ✓ | × | ✓ | ✓ | ✓ | × | × |
| NI-CT-FGSM | ✓ | ✓ | ✓ | ✓ | ✓ | × | × |
| MI-Admix-TI-DIM | ✓ | × | ✓ | ✓ | × | × | ✓ |
| VMI-CT-FGSM | ✓ | × | ✓ | ✓ | ✓ | ✓ | × |
| VNI-CT-FGSM | ✓ | ✓ | ✓ | ✓ | ✓ | ✓ | × |

## J   EXPERIMENTS WITH MORE DATASETS

To show the generalization capability of ILA-DA, we conduct experiments with two more datasets, namely CIFAR-10 and CIFAR-100 (Krizhevsky, 2012). For CIFAR-10, we follow Li et al. (2020b), which considers six models: VGG19 with batch normalization (Simonyan & Zisserman, 2015), Wide ResNet-28-10 (WRN) (Zagoruyko & Komodakis, 2016), ResNeXt-29 (ResNeXt) (Xie et al., 2017), DenseNet-190 (DenseNet) (Huang et al., 2017), PyramidNet[†] (Han et al., 2017) (with Shake-Drop (Yamada et al., 2018) and AutoAugment (Cubuk et al., 2019)), and also GDAS (Dong & Yang, 2019). The former four models are also selected to be used in the CIFAR-100 setup, with the model parameters obtained from a public repository[4]. For both datasets, we randomly sample 5000 images from the test set that are classified correctly by all four models and we pick VGG19 to be the source model. The experimental results on CIFAR-10 and CIFAR-100 are shown in Table 13.

For both CIFAR-10 and CIFAR-100, the attack success rates of the baselines are much higher than that of ImageNet. Although ILA-DA mostly attains the highest attack success rate among the baselines, we observe that the degree of improvement is not as high as that in Table 1. It may be because of the tiny size and low resolution ($32 \times 32$ for both CIFAR-10 and CIFAR-100) of the images, leading to a reduction in the effect of data augmentation.

Table 13: Attack success rates of CIFAR-10 adversarial examples on different models, generated from VGG19.

| Dataset | Method | VGG19* | WRN | ResNeXt | DenseNet | P.Net[†] | GDAS | Average |
|---|---|---|---|---|---|---|---|---|
| CIFAR-10 | I-FGSM | 99.3% | 57.2% | 57.2% | 53.6% | 12.5% | 40.2% | 41.1% |
| | + ILA | 99.4% | 90.5% | 91.5% | 88.9% | 39.5% | 76.0% | 77.3% |
| | + ILA++ | 99.5% | 90.7% | 92.4% | 89.1% | 41.3% | 77.1% | 78.1% |
| | + DI & MI | 94.8% | 72.8% | 71.4% | 70.0% | 29.3% | 57.5% | 60.2% |
| | + LinBP | **100.0%** | **92.4%** | **92.9%** | **90.5%** | 45.3% | 81.1% | 80.4% |
| | + ILA-DA | 97.3% | 92.0% | 92.7% | **90.5%** | **52.6%** | **81.6%** | **81.9%** |

| Dataset | Method | VGG19* | WRN | ResNeXt | DenseNet | Average |
|---|---|---|---|---|---|---|
| CIFAR-100 | I-FGSM | 98.8% | 38.9% | 31.9% | 33.8% | 50.9% |
| | + ILA | **98.9%** | 80.8% | 72.7% | 73.9% | 81.6% |
| | + ILA++ | 98.5% | 81.2% | 74.8% | 74.6% | 82.3% |
| | + DI & MI | 96.1% | 69.3% | 62.8% | 63.0% | 72.8% |
| | + LinBP | 98.9% | 75.2% | 69.7% | 70.5% | 78.6% |
| | + ILA-DA | 97.5% | **87.1%** | **81.9%** | **81.6%** | **87.0%** |

\* The source model used to generate the attack.

---

[4] https://github.com/bearpaw/pytorch-classification

## K    Experiments with Transformer-based Models

All the previous experiments are conducted on CNN models. Due to the rapid growth of vision transformers, evaluating the robustness of transformer-based models became an important topic. Therefore, we test the transferability of ILA-DA on eight transformer-based models, including Vit-B/16 (Dosovitskiy et al., 2021), DeiT-B (Touvron et al., 2021a), LeVit-256 (Graham et al., 2021), PiT-B (Heo et al., 2021), CaiT-S-24 (Touvron et al., 2021b), ConVit-B d'Ascoli et al. (2021), TNT-S (Han et al., 2021) and Visformer-S (Chen et al., 2021). We use ViT as the source model and report the transferred attack success rates in Table 14. The pre-trained weights are obtained from TIMM (Wightman, 2019). We apply ILA-DA to the output of the 8-th attention block.

Table 14 reflects a similar result to CNN models, i.e. ILA-DA exceeds the baselines by a large margin. In particular, ILA-DA acheives an average attack success rate of 56.3%, which is 4.2% higher than the VNI-CT-FGSM baseline. This shows that the proposed intermediate level attack with data augmentation can be generalized to transformer-based models.

Table 14: Attack success rates of 8 undefended transformers on ImageNet, generated from ViT with $\epsilon = 8/255(0.03)$.

| Method | ViT* | DeiT | LeViT | PiT | CaiT | ConViT | TNT | Visformer | Avg. |
|---|---|---|---|---|---|---|---|---|---|
| I-FGSM | 100.0% | 25.7% | 14.1% | 12.1% | 25.7% | 28.6% | 32.5% | 12.9% | 21.7% |
| MI-CT-FGSM | 89.3% | 43.3% | 36.1% | 29.8% | 44.7% | 47.0% | 59.1% | 33.3% | 47.8% |
| NI-CT-FGSM | 96.1% | 40.8% | 32.9% | 23.4% | 41.4% | 42.1% | 59.5% | 30.3% | 45.8% |
| VMI-CT-FGSM | 96.0% | **50.8%** | 35.7% | 29.4% | 49.9% | 50.9% | 63.1% | 34.1% | 51.2% |
| VNI-CT-FGSM | 96.9% | **50.8%** | 35.9% | 29.6% | **51.5%** | **52.1%** | 65.3% | 34.6% | 52.1% |
| ILA-DA (Ours) | **98.6%** | 48.3% | **49.3%** | **39.8%** | 50.3% | 49.6% | **71.2%** | **43.3%** | **56.3%** |

* The source model used to generate the attack.

## L    Experiments with Stronger Defended Methods

In Section 4.3, the candidate defended models are selected based on the submissions in NeurIPS 2017 Adversarial Competition. Considering the recent advancements in adversarial training, we test our method on five additional defended models according to the recent leader board of ImageNet against untargeted attack from RobustBench (Croce et al., 2021). The additional defended models include the ResNet50 and Wide-ResNet50 2x (WRN) models from Salman et al. (2020), the ConvNeXt (Liu et al., 2022) and XCiT-L12 (El-Nouby et al., 2021) models from Debenedetti et al. (2022) and the ResNet50 model from Wong et al. (2020). We pick ResNet50 as the source model to generate the ILA-DA attacks.

Similar to the experimental setup in Section 4.3, 1000 test images were used. However, different from the previous experiments, the models do not reach 100% clean accuracy on the samples. Hence, we only consider an attack to be successful if $F(\mathbf{x}) = y$ and $F(\mathbf{x}') \neq y$ for a model $F$ and ground-truth $y$. Note that this causes the number of effective samples evaluated on the models less than 1000. The result is shown in Table 15.

Table 15: Attack success rates of defended models on ImageNet, generated from undefended ResNet50 with $\epsilon = 16/255(0.063)$.

| Source Model | Salman et al. (2020) | | Debenedetti et al. (2022) | | Wong et al. (2020) | Average |
|---|---|---|---|---|---|---|
| | ResNet50 | WRN | ConvNeXt | XCiT | ResNet50 | |
| I-FGSM | 1.6% | 1.4% | 1.0% | 0.3% | 2.5% | 1.4% |
| MI-CT-FGSM | 11.0% | 11.1% | 9.0% | 7.4% | 15.0% | 10.7% |
| NI-CT-FGSM | 10.7% | 10.1% | 8.1% | 6.6% | 14.5% | 10.0% |
| VMI-CT-FGSM | 11.8% | 11.8% | 10.6% | 8.7% | 16.5% | 11.9% |
| VNI-CT-FGSM | 11.6% | 12.1% | 9.8% | 8.2% | 16.5% | 11.6% |
| ILA-DA (Ours) | **27.3%** | **24.5%** | **19.2%** | **19.3%** | **35.7%** | **25.2%** |

# M   EFFECT OF $\lambda$ AND THE REFERENCE ATTACK IN ATTACK INTERPOLATION

In this section, we first study the effect of $\lambda$ used for attack interpolation, with fixed values between 0 and 1, in addition to the adaptive choice computed from the norm of the feature map discrepancy. The result is shown in Figure 6. While a non-trivial value of $\lambda$ suffices to improve the performance, the adaptive selection using Equation (9) results in optimal or sub-optimal attack transferability in most cases.

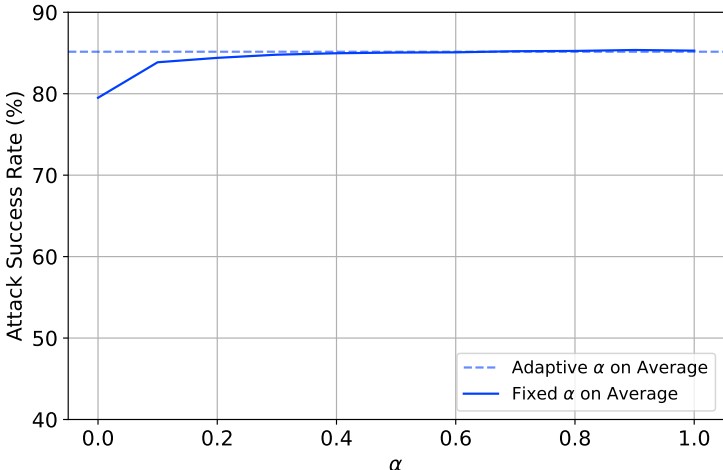

Figure 6: Average attack success rates over different values of $\alpha$, using ResNet50 as the source model with $\epsilon = 8/255$ (0.03).

As the reference attack $\mathbf{x}'$ is being interpolated during ILA updates, we visualize the attack transferability over different ILA iterations in Figure 7. In addition, We compare the effect of using different initial reference attacks, such as single-step FGSM, and an I-FGSM with less iterations. The result shows that a stronger initial reference attack in general leads to a more transferable fine-tuned attack.

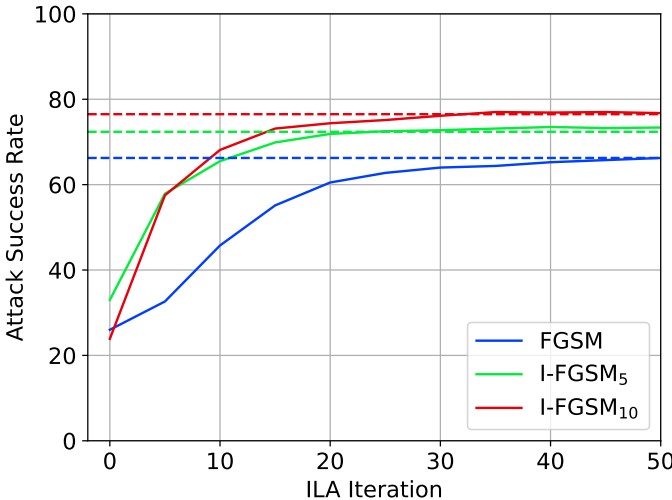

Figure 7: Attack success rates of the interpolated reference attack $\mathbf{x}'$ over the ILA iterations with different reference attacks. The dotted lines indicate the final attack success rate of the fine-tuned attacks $\mathbf{x}''$.

# N   STUDY ON THE RUNNING TIME OF THE ALGORITHMS

In this section, we compare the running time between ILA-DA and other state-of-the-art attacks. We report the average running time required to run attack on a batch from the ILSVRC2012 validation set, with a batch size of 100. The remaining setup is the same as the previous experiments, where all hyper-parameters are specified in Appendix F. We measure the running time required for attacking a batch and report the mean and standard deviation of the running time across 5 batches. All the experiments are performed on Nvidia A100 GPU. The result is summarized in Table 16.

As a fine-tuning strategy, ILA-DA strengthens the reference attacks and search for augmentation iteratively. In the algorithmic aspect, we discuss the run-time of ILA-DA in three parts. First, for the automated data augmentation, the transformation applied to $\mathbf{x}$ and $\mathbf{x}'$ has a negligible computational time. No extra forward or backward pass is required in an iteration since the gradient of $p_\alpha$ can be obtained through the backpropagation of the ILA projection loss. Second, the reverse update does not incur additional costs since $\mathbf{x}_{\text{rev}}$ can be computed arithmetically using $\mathbf{x}$ and $\mathbf{x}'$. Finally, for attack interpolation, $\Delta y_l''$ is obtained by a partial forward pass up to the selected intermediate layer. This is the major increase in computational cost of ILA-DA in comparison to the original ILA.

Although ILA-DA requires a slightly higher computational time (around 1 second) than ILA, the ILA-DA attack is much more transferable for both undefended and defended models (see Table 16). Apart from ILA-DA, other state-of-the-art attacks require comparably long running time to create highly transferable attacks. We also find that the methods with data augmentation, like MI-CT-FGSM and ILA-DA, benefit more from the increased number of iterations than those without data augmentation, like MI-FGSM. This verifies the use of data augmentation in transfer-based attacks.

Table 16: Running time comparison between ILA-DA and other attacks. The number in subscript indicates the number of iterations applied.

| Method | Running Time (s) | Average Attack Success Rate | |
| --- | --- | --- | --- |
| | | Undefended | Defended |
| MI-FGSM$_{10}$ | 1.1±0.03 | 63.1% | 8.6% |
| MI-FGSM$_{20}$ | 2.3±0.04 | 61.9% | 7.9% |
| MI-CT-FGSM$_{10}$ | 5.4±0.03 | 72.9% | 54.5% |
| MI-CT-FGSM$_{20}$ | 10.9±0.02 | 80.5% | 57.7% |
| VNI-CT-FGSM$_{10}$ | 33.4±1.39 | 77.8% | 71.5% |
| VNI-CT-FGSM$_{20}$ | 67.4±1.39 | 79.9% | 78.2% |
| LinBP$_{300}$ | 82.0±0.97 | 80.4% | - |
| ILA$_{50}$ | 6.1±0.01 | 65.0% | 6.6% |
| ILA$_{500}$ | 61.5±0.08 | 69.3% | 5.1% |
| ILA-DA$_{50}$ (Ours) | 7.6±0.07 | 84.5% | 38.8% |
| ILA-DA$_{200}$ (Ours) | 30.5±0.09 | 87.4% | 68.7% |
| ILA-DA$_{500}$ (Ours) | 74.9±0.12 | 88.7% | 81.1% |

## O   MORE VISUALIZATIONS ON THE GENERATED EXAMPLES

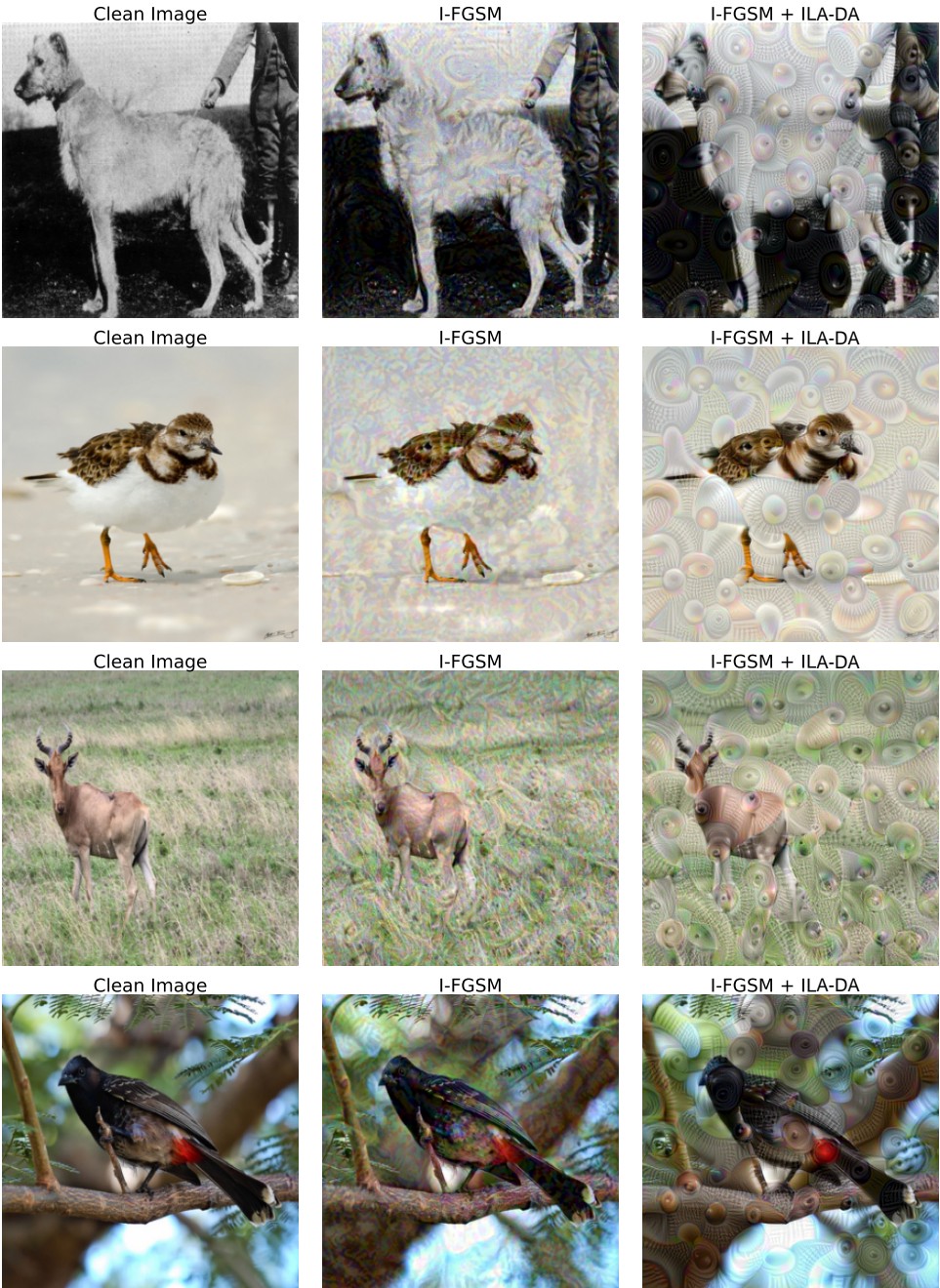

Figure 8: Visualization of the generated images among: clean image, ILA, and ILA-DA (Ours), with the $\epsilon = 16/255$ (0.063).

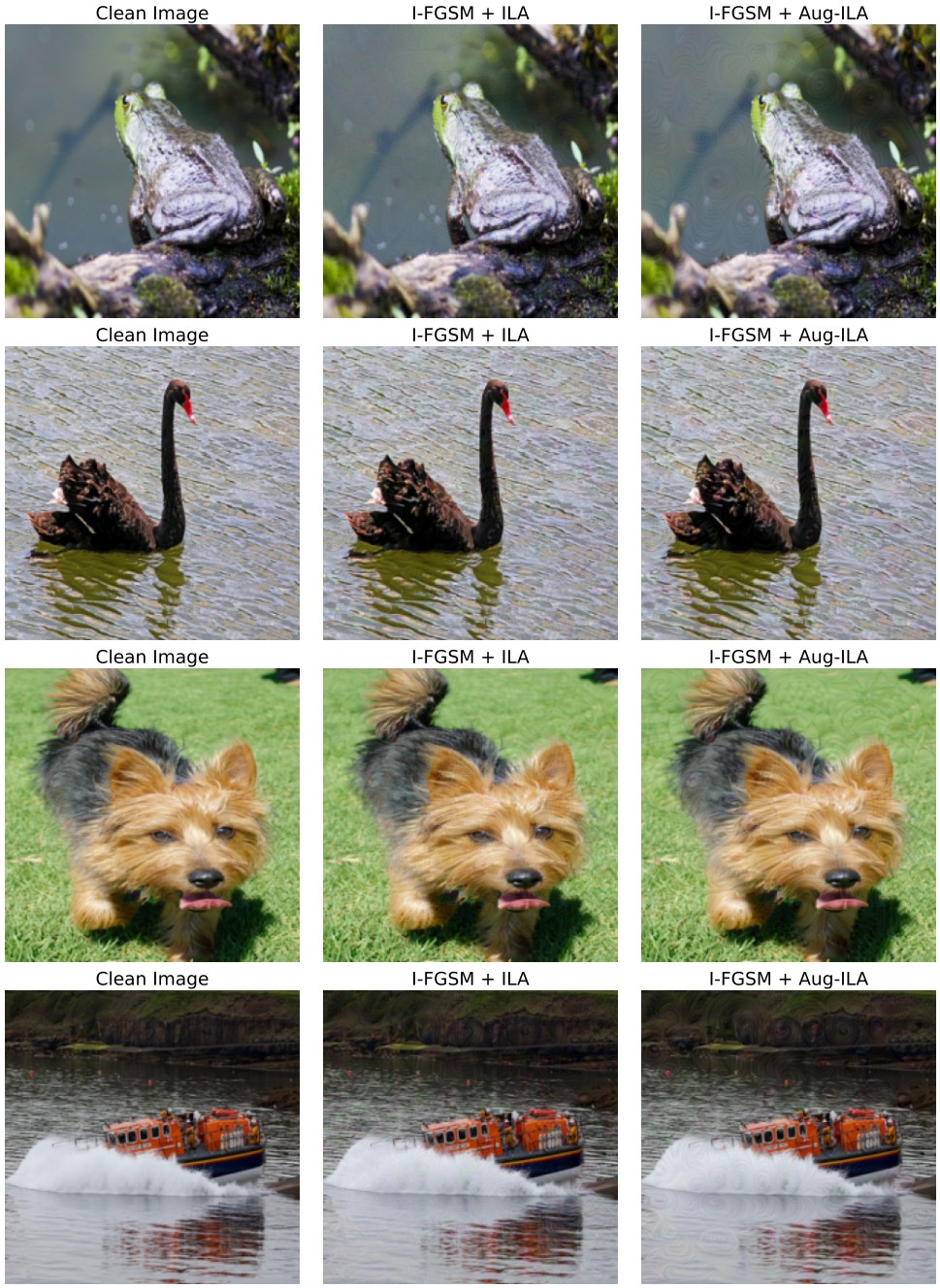

Figure 9: Visualization of the generated images among: clean image, ILA, and ILA-DA (Ours), with the $\epsilon = 5.0$ with $\ell_2$ norm constraint.

