# OpenReview forum: "ILA-DA: Improving Transferability of Intermediate Level Attack with Data Augmentation"
_ICLR.cc/2023/Conference — ICLR 2023 poster_

### Official Review · Reviewer_vZBZ · 2022-10-20

**Confidence:** 3
**Correctness:** 3
**Technical Novelty And Significance:** 3
**Empirical Novelty And Significance:** 2
**Recommendation:** 6

**Clarity, Quality, Novelty And Reproducibility:**

* Clarity: The paper is written well and the organization is good.

* Quality: The quality is good since extensive experiments showed promising results of the proposed LIA-DA, compared with other transfer-based attacks.

* Novelty: I think this paper is lack of novelty slightly.

* Reproducibility: This work was implemented on public datasets. The implementation details are detailed, although the author did not list it as an independent section. The reference code is provided, showing good reproducibility.

**Strength And Weaknesses:**

**Strength**

1. The structure of this paper is clear.
2. The experiments are conducted extensively on both undefended models and defended models,  and the proposed ILA-DA showed promising results comparing with other transfer-based attacks.
3. Ablation results demonstrate the effectiveness of each component.



**Weakness**
1. The choice of different intermediate layers may be important. Figure 3 shows that the performance fluctuates significantly when the intermediate layer is different. It means that when we have a new model, we have to spend much time to determine the optimal layer. However, other methods expect ILA have no such restriction. Can you comapre the results between ILA-DA with different intermediate layers and other transferable attack methods, like LinBP or CTM family? Maybe you can plot a figure like Figure. 3 in paper ILA[1].

2. In AutoAugment, the magnitude of augmentation may also influence the results. Do you consider it into the optimal augmentation choice?  In addition, did you try combine multiple transforms together to augment the data? Also, after learning the $p_{\alpha}$, will there be any trend in the data augmentation choices?

3. As for reverse adversarial update, the author expect that "the stronger the reference attack is, the better ILA performs", can you show the attack effectiveness of $x - (x - x^{'})$ comparing with other attack methods like FGSM?

**Reference**

[1] Qian Huang, Isay Katsman, Horace He, Zeqi Gu, Serge Belongie, and Ser-Nam Lim. Enhancing
adversarial example transferability with an intermediate level attack. In ICCV, 2019.

**Summary Of The Paper:**

This paper proposed some augmentation techniques to enhance the transferability of Intermediate-level Attacks(ILA), including automated data augmentation, reverse adversarial update, and attack interpolation. This paper evaluated proposed method on both undefended and defended models, showing the SOTA result comparing with some advanced baselines.

**Summary Of The Review:**

Generally, this paper proposes a simple and effective extension of the ILA. The extensive experiments on various benchmarks showed promising results for the proposed methods. However, this paper may have some problems with the sensitivity of the results to the choice of the intermediate layer.

---

> ### Author Response · Authors · 2022-11-17
> **Response to Reviewer vZBZ**
>
> We would like to thank the reviewer for the feedback and comments. Below are our responses and updates.
>
> > The choice of different intermediate layers may be important
>
> Admittedly, the choice of intermediate layers will influence the attack transferability. We have updated Figure 3 in Appendix B to compare the performance of ILA-DA using different intermediate layers with other baselines. Figure 3 shows that even with a sub-optimal choice of the intermediate layer, ILA-DA still outperforms other baselines. Besides, among all the attacks that require specifying an intermediate layer (e.g. ILA and LinBP), ILA-DA yields the highest transferability on every layer.
>
> We find that the choice of the intermediate layer is not sensitive to the victim models, i.e. the optimal intermediate layer discovered from one target model is likely to be a good choice to attack other target models. Therefore, it is not necessary to search for the best intermediate layer for every new victim model. Unlike hyper-parameters tuning in supervised learning tasks, generating a transfer-based attack generally has a faster feedback loop. Hence, searching for a good intermediate layer in ILA-DA does not take up significant computation power.
>
> > In AutoAugment, the magnitude of augmentation may also influence the results. Do you consider it into the optimal augmentation choice?
>
> Thanks for pointing this out. The magnitude of the transformation is set based on our preliminary findings. For example, random cropping with a ratio between 3%-10% (we later fixed it to be 5% in ILA-DA) works well on ImageNet. The performance gradually decreases when more than 10% of the image is cropped.
>
> > In addition, did you try combine multiple transforms together to augment the data? Also, after learning the Pα , will there be any trend in the data augmentation choices?
>
> In our ablation study, we find that combining three augmentations is less effective than using one augmentation at a time. We believe that using multiple augmentations may cause excessive distortions to the original reference attacks. In this case, optimizing the magnitude value may mitigate the problem. However, unlike AutoAugment which uses reinforcement learning, ILA-DA cannot optimize the augmentation magnitude directly if the operations are non-differentiable. Moreover, there exist multiple ways to combine the transformations, for example, AutoAugment uses a policy-sub-policy formulation to apply two transformations, while TANDA [1] applies multiple incremental transformations. The choice of the number of operations is also arbitrary. Therefore, we focus on optimizing single augmentations with fixed magnitude in ILA-DA and leave the research of learnable magnitudes and multiple transformations as future work.
>
> Also, we illustrate the learned parameter vector $p$ in Appendix C. We find that random cropping is more effective than translation and rotation. Among the data augmentation that we have tested, rotation is the least effective transformation.
>
> > As for reverse adversarial update, the author expect that "the stronger the reference attack is, the better ILA performs", can you show the attack effectiveness of  x−(x−x′) comparing with other attack methods like FGSM?
>
> The general assumption of  "the stronger the reference attack is, the better ILA performs" is the inspiration of attack interpolation which gradually strengthens the reference attack. The term $\textbf{x} - (\textbf{x}’ - \textbf{x})$ is indeed the example used for reverse update.
>
> From our understanding, we believe reviewer vZBZ would like to see verification on whether a strong reference attack improves the attack success rate after fine-tuning with attack interpolation. Therefore, we present an additional experiment to measure the attack transferability of the reference attack $\textbf{x}’$ during the ILA updates (see Figure 7 in Appendix M).
>
> In the figure, initially, a strong attack on the source model does not necessarily mean better transferability. However, with the tuning of attack interpolation, the strongest reference attack, I-FGSM10 yields the highest overall attack success rates among all tested attacks. Also, the fine-tuned attack that fine-tunes I-FGSM10 is the most transferable attack (as reflected by the dotted lines). This empirically verifies the intuition of the statement  "the stronger the reference attack is, the better ILA performs". We also tested I-FGSM with more iterations but the result is saturated and does not further give an apparent growth.
>
>
> ### References
>
> [1] A. J. Ratner, H. R. Ehrenberg, Z. Hussain, J. Dunnmon, and C. Re. Learning to compose domain-specific transformations for data augmentation. In NeurIPS, 2017.

---

> > ### Comment · Reviewer_vZBZ · 2022-12-05
> > **Thanks for the Response**
> >
> > Thanks for the response. My concern is cleared. Different choices of intermediate layers can all get good performance. The experiment is extensive and the performance is satisfactory. I will keep my score.

---

### Official Review · Reviewer_KZsc · 2022-10-21

**Confidence:** 4
**Correctness:** 3
**Technical Novelty And Significance:** 2
**Empirical Novelty And Significance:** 2
**Recommendation:** 5

**Clarity, Quality, Novelty And Reproducibility:**

The novel is considered limited since the general idea of using data augmentation to improve the black-box transferability has been widely studied before. Though new strategies have been proposed for a special kind of attacks, the novelty upon previous works is insufficient.

**Strength And Weaknesses:**

Strengths:

+ This paper is clearly written. It does a great effort to review previous attacks.
+ The evaluation results are comprehensive, covering a lot of attacks.

Weaknesses:

- The main contribution of this work is to introduce data augmentation strategies to intermediate-level attacks. The general idea has been widely studied. The paper proposes three new strategies, but they are not novel enough.
- The evaluation is insufficient in evaluation models. For undefended models, only CNN-based models are adopted. The authors should try transformer-based models. For defended models, only weak defenses are considered. There are new defenses on ImageNet with strong performance based on adversarial training. The authors should try these new defenses.
- Can the new strategy, especially automated image transformation, be applied to other transfer-based methods (e.g., MI-FGSM, DIM, etc.)?

**Summary Of The Paper:**

This paper proposes an intermediate-level attack method with three data augmentation strategies. They include automated search for image transformation, a reverse adversarial update strategy, and an attack interpolation method. The proposed method can achieve higher transferability of the generated adversarial examples. Experiments on the ImageNet show the improvements over previous attack methods against undefended and defended models.

**Summary Of The Review:**

Despite the writing clarity, the work lacks novelty and sufficient evaluation on other models.

---

> ### Author Response · Authors · 2022-11-17
> **Response to Reviewer KZsc (Part 1/2)**
>
> We would like to thank the reviewer for the comments. We hope to resolve your concerns with the responses below.
>
>
> > The main contribution of this work is to introduce data augmentation strategies to intermediate-level attacks. The general idea has been widely studied. The paper proposes three new strategies, but they are not novel enough
>
> We agree that the general idea of applying data augmentation to improve adversarial transferability is studied in previous works. In particular, TIM, DIM and SIM adapt simple heuristic data augmentations (e.g. scaling and translation) that are found to be effective in supervised learning to enhance adversarial attacks.
>
> In ILA-DA, we study more advanced data augmentation techniques that are built to deliver stronger attacks explicitly. For example, the automated augmentation optimizes the ILA loss, and attack interpolation updates an existing reference attack towards the direction of a more transferable attack. Therefore, the careful design of the automated and ‘adversarial-based’ augmentations is a novel method compared to conventional data augmentations. In addition, most existing works study the use of data augmentation for logit-level attacks, while in our case, the data augmentation is applied in intermediate-level attacks but never reaches the logits. One advantage of perturbing in such a way is to enforce the attack to be applied in the feature space, which might better alter the models’ perception of the examples.
>
> > For undefended models, only CNN-based models are adopted. The authors should try transformer-based models
>
> Initially, we followed the previous works to evaluate our methods on CNN models only. However, due to the increasing concern about the robustness of vision transformers, it would be beneficial to include the results of transform-based models. In particular, we use ViT as the source model and create adversarial examples to attack other transformer architectures, including DeiT, LeVit, PiT, CaiT, ConViT, TNT and Visformer. The selection of these eight transformers is also used in Wei et al. [1]. Our experiment shows that ILA-DA attains an average attack success rate of $56.3\\%$, which is $4.2\\%$ higher than VNI-CT-FGSM. The details can be found in Table 14 in the updated Appendix K.
>
> |              | ViT* | DeiT | LeViT | PiT  | CaiT | ConViT | TNT  | Visformer | Avg     |
> |--------------|------|------|-------|------|------|--------|------|-----------|---------|
> | I-FGSM       | 100  | 25.7 | 14.1  | 12.1 | 25.7 | 28.6   | 32.5 | 12.9      | 21.7    |
> | MI           | 100  | 48.3 | 27.8  | 22.5 | 47.7 | 51.3   | 51.1 | 26.3      | 46.875  |
> | MI-CTM       | 89.3 | 43.3 | 36.1  | 29.8 | 44.7 | 47     | 59.1 | 33.3      | 47.825  |
> | NI-CTM       | 96.1 | 40.8 | 32.9  | 23.4 | 41.4 | 42.1   | 59.5 | 30.3      | 45.8125 |
> | VMI-CTM      | 96   | 50.8 | 35.7  | 29.4 | 49.9 | 50.9   | 63.1 | 34.1      | 51.2375 |
> | VNI-CTM      | 96.9 | 50.8 | 35.9  | 29.6 | 51.5 | 52.1   | 65.3 | 34.6      | 52.0875 |
> | ILA-DA       | 98.6 | 48.3 | 49.3  | 39.8 | 50.3 | 49.6   | 71.2 | 43.3      | 56.3    |
> \* The source model used to generate the attack
>
> > For defended models, only weak defenses are considered. There are new defenses on ImageNet with strong performance based on adversarial training. The authors should try these new defenses.
>
> We consider the defense methods reported in the NIPS 2017 Adversarial and Defences Competition (https://arxiv.org/abs/1804.00097). All models have undergone either adversarial training or ensemble adversarial training. Although they may no longer be the strongest defenses, they are considered a common standard used for evaluating white-box or black-box attacks.
>
> To test our proposed attack on stronger defenses, we further select 3 more advanced and recent defenses based on the leaderboard of RobustBench [2], namely the following papers:
> A Light Recipe to Train Robust Vision Transformers [3]
> Do Adversarially Robust ImageNet Models Transfer Better? [4]
> Fast is better than free: Revisiting adversarial training [5]
> The details are reported in Appendix L. From the table, the models are much more robust against transfer-based attacks, but the relative performance between the attacks is similar to Table 2.
>
> | Model         | ResNet50 [4] | WRN [4]   | ConvNeXt [3] | XCiT [3]  | ResNet50 [5] |   Average   |
> |---------------|----------|--------|----------|--------|----------|---------|
> | I-FGSM        |    1.60% |  1.40% |    1.00% |  0.30% |    2.50% |   1.40% |
> | MI-CT-FGSM    |   11.00% | 11.10% |    9.00% |  7.40% |   15.00% |  10.70% |
> | NI-CT-FGSM    |   10.70% | 10.10% |    8.10% |  6.60% |   14.50% |  10.00% |
> | VMI-CT-FGSM   |   11.80% | 11.80% |   10.60% |  8.70% |   16.50% |  11.90% |
> | VNI-CT-FGSM   |   11.60% | 12.10% |    9.80% |  8.20% |   16.50% |  11.60% |
> | ILA-DA (Ours) |   27.30% | 24.50% |   19.20% | 19.30% |   35.70% |  25.20% |

---

> > ### Comment · Reviewer_KZsc · 2022-12-05
> > **Thanks for the response**
> >
> > Thanks for the detailed response. The additional experiments can help address some of my concerns. Given the current experiments, it is clearer that the proposed method is effective over the state-of-the-art although the novelty is still marginal. Therefore, I raise my rating to 5.
> >
> > The authors are also encouraged to provide more results on ensembling methods (attacking multiple models and comparing with baselines, the question is "can the method be adapted to multiple models?"), but this will not affect my final rating.

---

> ### Author Response · Authors · 2022-11-17
> **Response to Reviewer KZsc (Part 2/2)**
>
> > Can the new strategy, especially automated image transformation, be applied to other transfer-based methods (e.g., MI-FGSM, DIM, etc.)?
>
> We further performed a study on the influence of the proposed automated augmentation policy to other baseline attacks. The details can be found in the new Appendix G.
>
> To sum up the findings, the automated image transformation can be applied to other transfer-based methods that use no augmentation, such as MI-FGSM. However, for the other methods that already employ a fixed augmentation scheme, like DIM and CTM, further applying AutoDA brings limited effects. We believe the use of multiple transformations may cause excessive distortions to the images and information loss, thereby degrading the attack transferability.
>
> For reverse update and attack interpolation, they cannot be directly applied to other transfer-based methods as they explicitly require a reference attack to fine-tune. However, the ideas of reusing past adversarial examples and subtracting the perturbation to create a “reverse update” can still be adopted in general.
>
> ### References
>
> [1] Z. Wei, J. Chen, M. Goldblum, Z. Wu, T. Goldstein, and Y. Jiang. Towards transferable adversarial attacks on vision transformers. In AAAI, 2022.
>
> [2] F. Croce, M. Andriushchenko, V. Sehwag, E. Debenedetti, N. Flammarion, M. Chiang, P. Mittal, and M. Hein. RobustBench: a standardized adversarial robustness benchmark. In NeurIPS Track on Datasets and Benchmarks, 2021.
>
> [3] E. Debenedetti, V. Sehwag, and P. Mittal. A Light Recipe to Train Robust Vision Transformers. ArXiv, abs/2209.07399, 2022.
>
> [4] H. Salman, A. Ilyas, L. Engstrom, A. Kapoor, and A. Madry. Do Adversarially Robust ImageNet Models Transfer Better? In NeurIPS, 2020.
>
> [5] E. Wong, L. Rice, and J. Z. Kolter. Fast is better than free: Revisiting adversarial training. In ICLR, 2020.

---

### Official Review · Reviewer_CjSu · 2022-10-23

**Confidence:** 5
**Correctness:** 4
**Technical Novelty And Significance:** 3
**Empirical Novelty And Significance:** 3
**Recommendation:** 8

**Clarity, Quality, Novelty And Reproducibility:**

This paper provides novel methods on using reverse adversarial update and attack interpolation, and apply novel application to existing AutoAugment technique into ILA generation. The paper is clearly written with high quality.

**Strength And Weaknesses:**

## Strength
1. This paper provides novel and practicla techniques on improving the ILA transfer attack
2. Thorough ablation study is provided to show the effectiveness of each proposed technique
3. The proposed method is evaluate on multiple datasets and models, as well as in combination with different base attack methods, further proving the general effectiveness under different scenarios.

## Weakness
1. There are some previsou feature space attack techniques that are worth mentioning and comparing to [1,2]
2. To my understanding the propoaed attack may need additional iterations to search for data augmentation schemes and finetune the attack image. It would be good to discuss the cost of generating the proposed attack, how is it compared to baselines, and if baseline methods can be improved with more attack iterations

[1] https://openaccess.thecvf.com/content_CVPR_2019/papers/Inkawhich_Feature_Space_Perturbations_Yield_More_Transferable_Adversarial_Examples_CVPR_2019_paper.pdf

[2] https://arxiv.org/pdf/2004.12519.pdf

**Summary Of The Paper:**

This paper provides 3 novel techniques: automated data augmentation, reverse adversarial update, and attack interpolation to improve the transferability of ILA attack, which achieve promising performance against multiple defended and undefended models.

**Summary Of The Review:**

In summary I believe the proposed method is novel, practical, and effective. This method can be a valuabel contribution to the transfer attack community. Thus I would recommend acceptance.

---

> ### Author Response · Authors · 2022-11-17
> **Response to Reviewer CjSu**
>
> Thank you very much for agreeing on the effectiveness of our proposed method. Also, we appreciate the reviewer's suggestions very much. Below is our reply and update.
>
> > There are some previous feature space attack techniques that are worth mentioning and comparing to [1,2]
>
> Thank you for suggesting the related works. We have included the two works in Section 2.2. Although both works utilize intermediate features to generate adversarial attacks, AA/FDA directly generates the attack, while ILA fine-tunes a given reference attack. With the reference attack, we can apply the proposed reverse update and attack interpolation to create even stronger attack samples.
>
> >  It would be good to discuss the cost of generating the proposed attack, how is it compared to baselines…
>
> Thank you for the suggestion. We have revised Table 16 in Appendix N, which compares the running time between different methods. In the table, ILA-DA requires slightly more time (~1s) than ILA but the fine-tuned attack can be much more transferable. For attacking undefended models, ILA-DA has the best efficiency and effectiveness since 50 iterations of ILA-DA already suffice to achieve the highest attack success rate among all the methods. For defended models, ILA-DA has a comparable efficiency compared with SOTA like VNI-CT-FGSM.
>
> > if baseline methods can be improved with more attack iterations
>
> In the updated Table 16, it depends on whether data augmentation methods are applied in the attack generation. For example, although increasing the number of iterations in MI-FGSM makes stronger attacks on the source model, the attack transferability turns out to be worsened. On the other hand, with the aid of data augmentation, the CTM family exhibits higher transferability with more iterations. This observation is similar to and consistent with that of ILA-DA, verifying the advantages of applying data augmentation techniques.

---

> > ### Comment · Reviewer_CjSu · 2022-11-18
> > **Feedback on author response**
> >
> > I would like to thank the author for the response. My doubt on the paper is cleared. I also checked other reviews and think most of us believe the paper proposes an effective method with thorough experiments. As for novelty, though both ILA and augmentation has been explored previously, the ablation provided in Sec 4.4 shows that the novel techniques proposed in the paper brings nontrival improvements. I especially like the insight of improving feature space confidence with reverse adversarial update, which can be inspiring for future work. In this case I think this paper makes adequate contribution. I will keep my score and argue for acceptance.

---

### Official Review · Reviewer_weXS · 2022-10-25

**Confidence:** 3
**Correctness:** 3
**Technical Novelty And Significance:** 2
**Empirical Novelty And Significance:** 2
**Recommendation:** 5

**Clarity, Quality, Novelty And Reproducibility:**

Clarity: The paper is clear and easy to follow.

Quality: Quality is good.

Novelty: the novelty looks okay -- although it looks like combining two items together, the results seem good.

Reproducibility: should be able to reproduce, everything is clear.

**Strength And Weaknesses:**

Strength
1. The idea is straightforward and easy to follow. Since the high-level idea of ILA and DA are straightforward, I think readers can easily get the idea of ILA-DA, which combines them together. I also appreciate Algorithm 1, which helps people reimplement it.
2. The experimental results show the effectiveness of the proposed method.
3. The experiments (especially in appendix) are solid – helps readers to better understand the proposed method.

Question
1. The proposed data augmentation seems very effective. Wondering if it can only be applied to ILA? If my understanding is correct, the idea of the proposed augmentation is not specifically designed for ILA, and can also be applied to other transfer-based adversarial attacks?

Weakness
1. In figure 1, I see the perturbation generated by ILA+DA is more significant than I-FGSM noise. I assume this is because the human vision is not perfectly align with the perturbation budget ϵ = 16/255. If so, maybe it would be convincing to also test on some other perturbation budget pattern (eg, under L2 norm metric) that might align better with human vision.
2. Writing: although this paper is easy to follow, I think readers may misunderstand that the improvement is trivial because both ILA and DA improve the transferability. Actually, I think the data augmentation is interesting and can be applied to other frameworks as well. It could be helpful if the authors can write more interesting findings and insights.


**Summary Of The Paper:**

This paper proposes to improve the transferability of adversarial examples. Previous studies show (1) Intermediate Level Attack(ILA) is effective, which is to finetune an adversarial example to improve transferability; and (2) data augmentation (DA) can also improve the transferability. This paper proposes a novel method ILA-DA, that combines Intermediate Level Attack(ILA) and data augmentation (DA). The results outperform both baselines.


**Summary Of The Review:**

This project shows good results but, if I understand is correct, the story and the writing style make the paper less interesting. I would be good to further understand the proposed method and then adjust the story.

---

> ### Author Response · Authors · 2022-11-17
> **Response to Reviewer weXS (Part 1/2)**
>
> We would like to thank the reviewer for the comments and questions. Below is our reply and update.
>
> > The proposed data augmentation seems very effective. Wondering if it can only be applied to ILA? If my understanding is correct, the idea of the proposed augmentation is not specifically designed for ILA, and can also be applied to other transfer-based adversarial attacks?
>
> We conducted a study on the influence of the proposed automated data augmentation (Auto-DA) method on other adversarial attack frameworks. The details can be found in the new Appendix G.
>
> To sum up the findings, Auto-DA can bring significant improvements to most adversarial attack frameworks that use no data augmentation. For attacks that already employ certain transformations, such as the CTM family, Auto-DA does not bring significant effects or occasionally worsens the attack transferability. We believe this is because excessive distortion to the image results in information loss and hence degrades the transferability. Despite the massive improvement, none of the experimented baselines could outperform ILA-DA, verifying the effectiveness of combining ILA and Auto-DA.
>
> > In figure 1, I see the perturbation generated by ILA+DA is more significant than I-FGSM noise. I assume this is because the human vision is not perfectly align with the perturbation budget ϵ = 16/255. If so, maybe it would be convincing to also test on some other perturbation budget pattern (eg, under L2 norm metric) that might align better with human vision.
>
> Observed in the qualitative result, ILA-DA tends to perturb the image with a “DeepDream-like” pattern that consists of “circles” and “spirals”, which may be more noticeable by human vision compared with jittering noise. One possible interpretation is that the extensive augmentation and intermediate level loss encourage the attack to fool the perceptual components (such as feature extraction) of the model. As a result, the perturbation poses a more noticeable influence on human perception, even though all the attacks are conducted under the same perturbation size.
>
> We tested with $\epsilon = 5.0$ under L2 norm and obtained the following result (which is also added to Table 5 in Appendix D):
>
> |             | ResNet50* | Inc V3 | WRN   | VGG19 | PNASNet | DenseNet | ResNeXt | MobileNet | SENet | Average |
> |-------------|----------|--------|-------|-------|---------|----------|---------|-----------|-------|---------|
> | I-FGSM      | 99.98    | 15.04  | 38.44 | 26.06 | 17.12   | 29.52    | 27.7    | 30.58     | 41.46 | 28.24   |
> | ILA         | 99.98    | 17.86  | 57.72 | 42.24 | 23.58   | 44.2     | 45.18   | 44.42     | 59.32 | 41.815  |
> | ILA++       | 100      | 17.08  | 61.18 | 43.64 | 21.12   | 46.6     | 48.16   | 45.76     | 62.32 | 43.2325 |
> | LinBP       | 100      | 14     | 49.33 | 37.5  | 18.67   | 36.67    | 34.67   | 33.83     | 52.33 | 34.625  |
> | MI-CT-FGSM  | 92.7     | 43.1   | 55    | 47.5  | 47.8    | 56.4     | 45.5    | 53.5      | 61.7  | 51.3125 |
> | NI-CT-FGSM  | 93.9     | 34.7   | 41.2  | 39.8  | 42.6    | 41.6     | 31.4    | 43.2      | 49.2  | 40.4625 |
> | VMI-CT-FGSM | 95.4     | 42.5   | 52.4  | 46    | 46.2    | 52.9     | 42      | 51.3      | 59.2  | 49.0625 |
> | VNI-CT-FGSM | 96.9     | 37.8   | 47.4  | 42    | 43.4    | 47       | 35.2    | 47.4      | 52.7  | 44.1125 |
> | ILA-DA      | 96.32    | 33.96  | 69.6  | 68.4  | 42.2    | 62.4     | 58.76   | 61.76     | 69.64 | 58.34   |
>
> In the table above, we observe similar findings to that of L-inf norm, that ILA-DA could still achieve the best attack success rate among all the methods.
>
> Moreover, we have also added a qualitative visualization of the attack under L2 norm constraint to Appendix N. If we zoom in closely, the “DeepDream-like” patterns can still be perceived.

---

> ### Author Response · Authors · 2022-11-17
> **Response to Reviewer weXS (Part 2/2)**
>
> > Writing: although this paper is easy to follow, I think readers may misunderstand that the improvement is trivial because both ILA and DA improve the transferability. Actually, I think the data augmentation is interesting and can be applied to other frameworks as well. It could be helpful if the authors can write more interesting findings and insights.
>
> Apart from conventional data augmentation, the ILA framework enables the use of “adversarial-based” augmentation” which utilizes the existing adversarial attack as a reference. For example, attack interpolation updates the reference attack based on the fine-tuned attack. Similarly, reverse adversarial update perturbs the clean example such that the source model produces different intermediate features. These operations can be regarded as strengthening an adversarial example with another adversarial example, which is quite different from conventional data augmentation techniques. More importantly, due to the ILA framework, the reference attack is ready as an input. We do not need to generate an extra attack in order to apply the “adversarial-based” augmentation”. This enables us to further improve the attack transferability without much computational overhead.
>
> Therefore, we believe combining ILA and DA can yield extra attack transferability, rather than a trivial combination of two incremental ideas. In the meantime, we also discover the potential of an “adversarial-based” augmentation in the fine-tuning phase, which was never considered in previous works.
>
> Another interesting finding is relevant to weakness (1). As discussed above, the attacks generated by combining “ILA” and “DA” exhibit a pattern with circles and spirals. This pattern, however, cannot be observed when ILA and DA are applied separately. Provided that ILA-DA achieves the best attack transferability among all of the studied works and also the patterns tend to affect human vision, one possible future direction in this area might be to target the perceptual components of the model, instead of its loss at the output.
>
> We hope the above response could resolve the reviewer's concern and deliver interesting findings.

---

> > ### Comment · Reviewer_weXS · 2022-11-19
> > **Thanks for the Response!**
> >
> > After carefully reading this paper and the comments from other reviewers, I tend to agree that there are interesting findings. However, I still feel the presentation of this paper could be better. For example, according to Figure 1, people may think the perturbation of ILA-DA is larger    in the perspective of human vision (maybe also present a figure from Figure 9?). Besides, I also feel the name of the proposed method "ILA-DA" is misleading -- readers may feel you just combine with two techniques together in a simple way... To summarize, I think the presentation of this paper can be improved.

---

### Author Response · Authors · 2022-11-17
**The Summary of Revision**

We would like to thank all the reviewers for their constructive feedback and suggestions. To improve the paper, we have made the following amendments:
- Section 2.2: More related works are added and discussed.
- Appendix B: Figure 3 is updated, with a clearer scale and more baselines (as horizontal dotted lines). Besides, for consistency, we have excluded the source model when computing the average attack success rate.
- Appendix D: A new table showing the attack success rate under the L2 norm constraint.
- [New] Appendix G: New experiments on the effect of the proposed automated data augmentation policy on the baselines.
- [New] Appendix K: New experiments with transformer-based models.
- [New] Appendix L: New experiments with more and stronger defended models.
- Appendix M: A new figure showing the effect of the strength of the reference attack.
- Appendix N: The running time comparison (Table 16) is updated, with more baselines and findings.
- Appendix O: Additional visualization with generated attacks under L2 norm constraint.
- Minor adjustments to the phrasing and format in the manuscript.

Note: All the changes are highlighted in blue.

We hope our modifications could resolve the concerns raised by the reviewers. Once again, we would like to thank all the reviewers and the ACs for reviewing the paper.

---

### Decision · Program_Chairs · 2023-01-20

**Decision:**

Accept: poster

**Justification For Why Not Higher Score:**

Though very strong empirical results are provided, the technical contribution of this paper is not particularly significant. As a result, we believe that accepting the paper as a "poster" is an appropriate decision.

**Justification For Why Not Lower Score:**

This paper conducted an interesting study on transferable adversarial examples, and obtained strong results. Despite the lack of a strong technical contribution, we believe that the paper is still worthy of being presented to the general ICLR audience.


**Metareview: Summary, Strengths And Weaknesses:**

This paper studies transferable adversarial examples. By building upon the prior work Intermediate Level Attack (ILA), this paper developed three novel augmentation strategies to boost the transferability of adversarial examples further. Experiments on ImageNet are provided to demonstrate its effectiveness.

Overall, the reviewers appreciate the simplicity and effectiveness of the proposed method, but have several major concerns regarding its empirical evaluations: 1) only weak defenses are considered; 2) no analysis of computational cost; 3) it is unclear whether the compared baselines are strong attackers. These concerns all get cleared in the rebuttal and the AC-reviewer meeting. As a result, our final decision is to accept this paper.

The AC agrees that including the additional results of ensemble attacks (as suggested by Reviewer KZsc) can further improve the quality and completeness of this paper.




**Note From Pc:**

if the above contains the word "oral" or "spotlight" please see: "oral" presentation means -> notable-top-5% and "spotlight" means -> notable-top-25%. As stated in our emails, we are disassociating presentation type from AC recommendations

**Summary Of Ac-Reviewer Meeting:**

Two major concerns are discussed: 1) the computational cost of the proposed method, and 2) whether strong attack baselines are considered.

The first concern is addressed in the revised version of the paper, i.e., Table 16 shows that this method is more efficient than others.

The second concern relates to the setting of the paper, which focuses on attacking a single model rather than an ensemble of models. While it is true that attacking a single model is typically weaker than attacking an ensemble, we believe attacking a single model alone is still a legitimate setting, and that this paper provides a fair and comprehensive comparison to previous state-of-the-art attackers in this setting.

Given all concerns are well solved, we reached an agreement to accept this paper.